# Pervasive diffusion of climate signals recorded in ice-vein ionic impurities

Felix S. L. Ng[1]

[1]Department of Geography, University of Sheffield, Sheffield, UK

*Correspondence to*: Felix Ng (f.ng@sheffield.ac.uk)

**Abstract.** A theory of vein impurity transport conceived two decades ago predicts that signals in the bulk concentration of soluble ions in ice migrate under a temperature gradient. If valid, it would mean that some palaeoclimatic signals deep in ice cores (signals from vein impurities as opposed to matrix/grain-boundary impurities) suffer displacements that upset their dating and alignment with other proxies. We revisit the vein physical interactions to find that a strong diffusion acts on such signals. It arises because the Gibbs–Thomson effect, which the original theory neglected, perturbs the impurity concentration of the vein water wherever the bulk impurity concentration carries a signal. Thus, any migrating vein signals will not survive into deep ice where their displacement matters, and the palaeoclimatic concern posed by the original theory no longer stands. Simulations with signal peaks introduced in shallow ice at the GRIP and EPICA Dome C ice-core sites, ignoring spatial fluctuations of the ice grain size, confirm that rapid damping and broadening eradicates the peaks by two-thirds way down the ice column. Artificially reducing the solute diffusivity in water (to mimic partially-connected veins) by $10^3$ times or more is necessary for signals to penetrate into the lowest several hundred metres with minimal amplitude loss. Simulations incorporating grain-size fluctuations on the decimetre scale show that these can cause the formation of new, non-migrating solute peaks. The deep solute peaks observed in ice cores can only be explained by widespread vein disconnection or a dominance of matrix/grain-boundary impurities at depth (including their recent transfer to veins) or signal formation induced by grain-size fluctuations; in all cases, the deep peaks would not have displaced far. Disentangling the different signal contributions – from veins, the ice matrix, grain boundaries, and grain-size fluctuations – will aid robust reconstruction from ion records.

## 1 Introduction

Chemical impurity concentrations in ice cores yield diverse palaeoclimatic information (e.g. Legrand and Mayewski, 1997; Wolff et al., 2006). As with other ice-core proxies, such as stable water isotopes, it is generally hoped that post-depositional modification of signals in such records, which may hamper their interpretation, is minimal. For signals of abrupt or discrete (e.g., volcanic) events, of interest is whether their position and shape record their timing and magnitude faithfully. If, for instance, signals diffuse in the ice, neighbouring peaks may merge as they descend towards the ice-sheet base. Conceivably, a range of physico-chemical processes may distort signals, limiting the resolution and accuracy of the retrievable information.

In a landmark paper, Rempel et al. (2001) proposed a theory to show that signals in the bulk concentration of dissolved ionic impurities – major ions such as $SO_4^{2-}$, $Cl^-$ and $Na^+$ – in the water veins at polycrystalline grain junctions (Nye, 1989; Mader, 1992a, b) migrate relative to the ice when a temperature gradient is present. Driven by what they term "anomalous diffusion", migration occurs in the direction of rising temperature and displaces signals with minimal distortion, so their apparent age deviates from their true age. Rempel et al. (2001) calculated cumulative signal displacements of ~ 0.1–1 m in the lowest kilometre of the GRIP ice core. Their mechanism could decouple the ion records from other ice-core proxies and cause significant age errors in palaeoclimatic histories, especially in deep ice where temperature increases markedly towards the bed. This has prompted evaluation for signs of anomalous diffusion in some ice-core records (e.g. Tison et al., 2015) and simulation of the migration of specific species (e.g. methane sulphonic acid in firn; Osman et al., 2017). Note that it is non-trivial to infer the absence or amount of migration by comparing the records of signal peaks from different cores, due to uncertainty in depth-age scales, a lack of rigorous independent control of where individual peaks should lie, and spatial inhomogeneity in the atmospheric dispersal and deposition of impurities by environmental events. Some studies contend that signal migration may be limited – and thus the ion records dependable, because solute transport is hindered by disconnections in the vein network (Barnes et al., 2003; Barnes and Wolff, 2004) or because most ionic impurities are located at grain boundaries (Barnes and Wolff, 2004) or in salt micro-inclusions within ice crystals (Ohno et al., 2005), instead of in veins. How extensively ice-core chemical records have been altered by anomalous diffusion is unresolved, despite the relevance of this question to the synchronisation of ice-core age scales (e.g. Severi et al., 2007; Fujita et al., 2015) and dating of palaeoclimatic events.

Here we re-examine "Rempel's theory" of vein impurity transport (i.e. Rempel et al., 2001), discovering missing elements in it that change the predicted signal evolution. We find that the relevant processes cause *diffusion* of signals at rates that threaten their survival in ice cores, whether or not a temperature gradient drives their migration. Hence the theory's implications are radically revised. Our purpose is not to develop the theory of migrating signals to show how it can match observed ion records or be used in reconstructions, but to highlight the contrary: it struggles to explain the presence of distinct ionic peaks in deep ice. Figure 1 exemplifies such peaks in ice cores from Antarctica and Greenland. Much of our analytical and numerical work (Sects. 2 & 3) is spent on understanding the origin of the strong diffusion and showing its operation. We also correct the signal migration speed, and show that grain-size fluctuations can cause new impurity signals to form. At the end (Sect. 4) we discuss what the revised theory means for the provenance of the deep ionic peaks in ice-core records.

Following Rempel's framework, we model processes below the firn-ice transition and one ion species only ($SO_4^{2-}$ is used in our calculations). We let $c_B$ denote its bulk concentration, using the unit mol/L or M to refer to the amount of impurity in a unit volume of ice. Note that $c_B$ accounts only for impurities dissolved in the water veins and excludes impurities at grain boundaries and in the ice matrix (inside grains), which are not modelled. Chemical alteration of signals via cation–anion associations (e.g. Iizuka et al., 2004; Traversi et al., 2009), reaction of vein impurities with dust (Barnes and Wolff, 2004), the segregation of impurities to locations outside the veins and thermodynamic coupling between multiple ion species (Rempel et al., 2002; Rempel and Wettlaufer, 2003), are ignored.

Before plunging into mathematics, we first outline our main finding; see Fig. 2a–d, where $z$ denotes depth. In Rempel's theory, a centimetre/decimetre-scale peak in the bulk solute concentration $c_B$, representing a climate signal, is mirrored by variations in the porosity $\phi$ (Fig. 2c, d), which represents the volume fraction of veins in the ice. With $c_B$ encapsulating vein impurities, the relation $c_B = c\phi$ holds[1], where $c$, the solute concentration of the vein water, is determined by temperature $T$ through the liquidus. A background temperature gradient in the ice sets up a gentle gradient in $c$, driving downward solute diffusion through the vein network (Fig. 2a). Interestingly, the porosity modulates the solute diffusion flux so that on the trailing (upper) edge of the signal peak, $d\phi/dz > 0$ increases this flux with distance to draw down $c_B$, whereas on the leading (lower) edge of the peak, $d\phi/dz < 0$ reduces the flux with distance to bump up $c_B$. Thus the peak signal in $c_B$ translates (the same translation applies to trough signals). This mechanism is "anomalous" because *solute diffusion* causes the signal to move *without signal diffusion*, i.e., without changing its shape. In their calculations, Rempel et al. (2001) neglected the small effect of the vein-face curvature on the melting point (the Gibbs–Thomson effect), and a detailed justification of why this interfacial effect is small is given in their companion papers (Rempel et al., 2002; Rempel and Wettlaufer, 2003). Then $c$ depends on $T$ only, not on $c_B$. However, we find that when this seemingly reasonable approximation is not made, the $c_B$ signal causes a perturbation on $c(z)$ that drives non-negligible solute diffusion away from the peak (Fig. 2b), owing to the short length scale of the signal. Consequently, the $c_B$ peak experiences pronounced broadening and amplitude reduction. In an ice core, these conspire with vertical compression to regulate the evolving peak shape.

Besides Rempel et al. (2001), Barnes et al. (2003) have modelled vein-mediated evolution of dissolved ion signals below the firn-ice transition. Their main interest was to explain the signal diffusion found in the top 350 m of the EPICA Dome C ice core (EPICA community members, 2004), which they inferred from observed trends of peak broadening and damping in the sulphate and chloride records with age. To explain the signal diffusivities estimated for these ions – respectively, $4.7 \times 10^{-8}$ $m^2$ $yr^{-1}$ and $2 \times 10^{-7}$ $m^2$ $yr^{-1}$, they conceived two models of vein solute transport driven by grain growth, motivated by the fact that the mean grain size in their stretch of the core increases with depth. One model invokes local gradients in $c$ induced by porosity change during spatially non-uniform normal grain growth; this mechanism requires the presence of grain-size variations at the length scale of the $c_B$ signals. The other, an analogue model, uses a cell-based simulation to demonstrate how the disconnection of veins by grain growth (modelled as random removal of cells, and relocation of their impurity to neighbouring cells) causes diffusion even when the vein network is only partially connected. Both models predict no diffusion of $c_B$ unless there is net grain growth.

We do not incorporate the Barnes et al. (2003) mechanisms into our model here, as the "Gibbs–Thomson diffusion" is much faster than their diffusion (by an order of magnitude at least), and adding the latter strengthens our conclusions. Our

---

[1] Rempel et al. (2001) formulated their model with $\phi$ as a mass fraction (instead of volume fraction) and used a correspondingly different unit for $c_B$, but the interactions are the same.

mechanism is independent of grain growth and occurs in the absence of grain-size variations. We do, however, account for the continual motion of veins during grain boundary migration, which causes a slow "residual diffusion" on $c_B$. This mechanism

does not require a temperature gradient or grain-size variations to operate, and is unaffected by vein disconnection.

## 2    The model

### 2.1    Key relationships

We treat polycrystalline ice in a continuum description with variables as functions of position $\mathbf{x} = (x, y, z)$ and time $t$. Ice with

the mean grain size $d_g$ (grain diameter) has the vein length density

$$l \approx \frac{3}{d_g^2} \; ; \tag{1}$$

we adopt equality in this expression herein. If the vein faces have the radius of curvature $r_v$ (Fig. 2e; Nye, 1989), then the ice porosity $\phi$ is

$$\phi = \alpha r_v^2 l \, , \tag{2}$$

where $\alpha = 0.0725$ is a geometrical factor (Nye, 1991; Mader, 1992a, b). Recalling the relation between the bulk solute concentration and the solute concentration of the vein water,

$$c_B = \phi c \, , \tag{3}$$

and using (1) and (2), leads to

$$\frac{d_g^2 c_B}{3\alpha} = c r_v^2 \, . \tag{4}$$

For ice at a temperature $T$ below the reference temperature $T_0$, thermodynamic equilibrium between ice and vein water means that the liquidus relation is satisfied:

$$T_0 - T = \theta_c + \frac{\gamma T_0}{\rho_i L r_v} + \theta_p \, . \tag{5}$$

This equation features in earlier studies of the subject (Nye, 1991; Mader, 1992b; Rempel et al., 2001, 2002; Rempel and Wettlaufer, 2003; Barnes et al., 2003; Dani et al., 2012). The three terms on its right-hand side describe the temperature

depressions due to (i) solute, (ii) interfacial curvature (the Gibbs–Thomson effect) and (iii) pressure, respectively; $\rho_i$ is ice density, $L$ is latent heat, and $\gamma$ is the interfacial energy. Table 1 lists the constants used in this paper, and Table 2 lists our

model variables. For the first term in (5), a linear approximation $\theta_c = \Gamma c$ is valid at temperatures not far below the melting point, e.g. $\Gamma = 4.53\ \text{K M}^{-1}$ for $SO_4^{2-}$. The third term is typically small and may be absorbed into $T_0$ by accounting for glaciostatic overburden. Accordingly, the version of (5) that we use for the present analysis is

$$T_0 - T = \Gamma c + \frac{\gamma T_0}{\rho_i L r_v}, \tag{6}$$

but our simulations in Sect. 3 for specific ice-core sites will use (5), together with detailed nonlinear empirical formulas for $\theta_c$ and $\theta_p$ (Appendix A).

Given $d_g$, $T$ and $c_B$, (4) and (6) can be solved for $r_v$ and $c$ (e.g. Barnes et al., 2003). Figure 2f illustrates the solution as the intersection point of two curves. In their theory, Rempel et al. (2001) employed the liquidus relation without including the Gibbs–Thomson term, and Rempel et al. (2002) and Rempel and Wettlaufer (2003) argued that under glaciological conditions, a large $r_v$ makes the Gibbs–Thomson term negligible so that $\Gamma c \approx T_0 - T$; then $c(z)$ is dictated by the ice temperature profile (Fig. 2a). This approximation amounts to taking the intersection point in Fig. 2f to lie on the dashed line. However, as shown by Fig. 2f, the exact solution for $c$ does depend on $c_B$ – albeit weakly – for fixed $T$ and $d_g$. This dependence lies at the heart of our "Gibbs–Thomson diffusion". Specifically, when $\gamma T_0/\rho_i L r_v \ll \Gamma c$, a first-order approximate solution of (4) and (6) is

$$\Gamma c \approx (T_0 - T) - \frac{\gamma T_0}{\rho_i L r_v}, \quad \text{where} \quad r_v \approx \sqrt{\frac{c_B d_g^{\,2} \Gamma}{3\alpha(T_0 - T)}}, \tag{7}$$

or

$$c \approx \frac{1}{\Gamma}\left[(T_0 - T) - \frac{\gamma T_0}{\rho_i L}\sqrt{\frac{3\alpha(T_0 - T)}{d_g^{\,2}\Gamma}}\,c_B^{-1/2}\right]. \tag{8}$$

This differs from Rempel's approximation by the $\gamma$ (Gibbs–Thomson) term, which causes $\partial c/\partial c_B > 0$ and a perturbation to appear on $c(z)$ where $c_B(z)$ exhibits a signal (Fig. 2b, d). The rate of signal diffusion stemming from this minute perturbation will be quantified later. The result in (8) also shows that fluctuations in $d_g$ and $T$ could perturb $c(z)$ to drive solute transport and influence $c_B$. We will explore the potential effect of grain-size fluctuations (Sect. 3.4), but not temperature fluctuations, as these decay quickly in ice.

If, instead of (6), the full liquidus relation (5) is analysed, with $\theta_c = f(c)$ being a mildly nonlinear function, then the above findings are qualitatively unchanged, and (8) would read $c \approx f^{-1}([\ ])$ with [ ] as given in (8) except that $f^{-1}(T_0 - T)$ replaces $(T_0 - T)/\Gamma$ in the square root.

## 2.2 Porosity, water and solute conservation

The rest of the model is now formulated, for ice deforming with velocity $\mathbf{u} = (u, v, w)$. As the porosity is very small, $\phi \sim 10^{-6}$ << 1, the incompressibility condition $\nabla.\mathbf{u} = 0$ holds. The total transport flux of porosity is $\mathbf{u}\phi - \kappa\nabla\phi$. The first term here describes advection by ice. The second (Fickian) term describes a net transport of porosity due to the random, unceasing vein motion that accompanies grain boundary migration. We detail its physical derivation in Appendix B. The diffusivity $\kappa$ is given by

$$\kappa = \frac{K(T)}{3c_1}, \tag{9}$$

where $K = K_0\exp(-Q/RT)$ is the temperature-dependent grain growth rate, $R$ is the gas constant, $Q$ is activation energy, and $c_1 \approx 2$ to $3$ (we use 2.5 in our simulations).

Accordingly, porosity conservation is described by

$$\frac{\partial\phi}{\partial t} + \nabla.(\mathbf{u}\phi - \kappa\nabla\phi) = \frac{m}{\rho_i}, \tag{10}$$

where the melt rate $m$ accounts for phase change occurring at the interfacial boundaries of the veins; $m$ is the rate of mass melted per unit volume of ice. This result can also be derived from ice mass conservation.

If $\mathbf{q}$ is the water flux percolating through the vein network, then water mass conservation requires

$$\rho_w\left[\frac{\partial\phi}{\partial t} + \nabla.(\mathbf{q} + \mathbf{u}\phi - \kappa\nabla\phi)\right] = m, \tag{11}$$

and we deduce from this together with (10) that

$$\nabla.\mathbf{q} = -(1-r)\frac{m}{\rho_i}, \tag{12}$$

where $r = \rho_i/\rho_w \approx 0.92$ is the ratio of ice density to water density.

The local solute transport flux is $\mathbf{u}c_B + \mathbf{q}c - D\phi\nabla c - \kappa\nabla c_B$. This describes the summed effects of advection by ice and water flow, molecular diffusion in vein water, and random vein motion (as that driving porosity diffusion; Appendix B); $D$ is the diffusion coefficient for the ionic impurity in water, and the diffusivity $\kappa$ was defined in (9). The corresponding solute conservation equation is

$$\frac{\partial c_B}{\partial t} + \nabla.(\mathbf{u}c_B + \mathbf{q}c) = \nabla.(\phi D\nabla c + \kappa\nabla c_B). \tag{13}$$

Following Rempel's theory, we define the "anomalous velocity"

$$\mathbf{u}_c = -D\frac{\nabla c}{c} \tag{14}$$

and use $c_B = \phi c$ to rewrite the $D$-contribution in (13) to derive

$$\frac{\partial c_B}{\partial t} + \nabla.[(\mathbf{u}+\mathbf{u}_c)c_B + \mathbf{q}c] = \nabla.(\kappa\nabla c_B) . \tag{15}$$

If both water flux $\mathbf{q}$ and the $\kappa$-term ("residual diffusion" due to vein motion) are ignored, this simplifies, after using the

incompressibility condition $\nabla.\mathbf{u} = 0$, to the Rempel et al. (2001) equation:

$$\frac{\partial c_B}{\partial t} + (\mathbf{u}+\mathbf{u}_c).\nabla c_B = -(\nabla.\mathbf{u}_c)c_B . \tag{16}$$

As is well rehearsed in their theory, the advection term here predicts migration of chemical signals at the velocity $\mathbf{u}_c$ relative to the ice, with $\mathbf{u}_c$ controlled by the temperature profile via the liquidus and (14). Our analyses are to reveal departures from these predictions.

## 2.3    One-dimensional model

For an ice core beneath a flow divide or summit, where the ice motion is downward ($\mathbf{u} = (0, 0, w)$) and horizontal variations in $\phi$ and $c_B$ are negligible, (10) and (15) become (on using $\nabla.\mathbf{u} = 0$ again)

$$\frac{\partial\phi}{\partial t} + w\frac{\partial\phi}{\partial z} = \frac{m}{\rho_i} + \frac{\partial}{\partial z}\left(\kappa\frac{\partial\phi}{\partial z}\right) , \tag{17}$$

$$\frac{\partial c_B}{\partial t} + (w+w_c)\frac{\partial c_B}{\partial z} + \frac{\partial}{\partial z}(qc) = \frac{\partial}{\partial z}\left(\kappa\frac{\partial c_B}{\partial z}\right) - c_B\frac{\partial w_c}{\partial z} , \tag{18}$$

where $w_c = -(D/c)\partial c/\partial z$ is the anomalous velocity downward. (18) tracks the evolution of the bulk impurity profile $c_B(z, t)$ and its signals, but the water flux $q$ needs to be found via

$$\frac{\partial q}{\partial z} = -(1-r)\frac{m}{\rho_i} , \tag{19}$$

with $m$ calculated from (17). This problem is supplemented by (4) and (5) (or (6)), which give the instantaneous distributions

of $c$ and $\phi$ from the depth profiles of $c_B$, $T$ and $d_g$. There are five equations ((4), (5), (17)–(19)) for the five unknowns $c_B$, $c$, $\phi$,

$q$ and $m$.

## 2.4 Correction to the signal migration velocity

Here we explain the first departure from Rempel's original predictions: the migration speed will be faster than $w_c$, because of water percolation induced by the migration. To see this, we need to put all advective parts of (18) (notably $\partial(qc)/\partial z$) in terms of $c_B$. Let $c(z)$ be time-invariant as in Rempel's theory, and ignore the $\kappa$-term (vein motion), to write (18) as

$$c\left(\frac{\partial\phi}{\partial t}+\frac{\partial q}{\partial z}\right)+(w+w_c)\frac{\partial c_B}{\partial z}=-c_B\frac{\partial w_c}{\partial z}-q\frac{\partial c}{\partial z} \ .$$

Substituting for $\partial q/\partial z$ from (19), using (17) for $m$, converts this to

$$r\left(\frac{\partial c_B}{\partial t}+w\frac{\partial c_B}{\partial z}\right)+w_c\frac{\partial c_B}{\partial z}=-c_B\frac{\partial w_c}{\partial z}-q\frac{\partial c}{\partial z}-(1-r)w\phi\frac{\partial c}{\partial z} \ . \tag{20}$$

The $z$-derivatives on the left-hand side describe signal advection by ice flow and anomalous diffusion, respectively. Following the scaling argument in the Supplementary Information of Rempel et al. (2001), all terms on the right, based on the background gradient in $c$ (Fig. 2a), are negligible on the length scale of climatic signals ($\lesssim 10^{-1}$ m). In (20), the prefactor $r$ of the time derivative means that the migration speed is $w_c/r$, which exceeds $w_c$ by $\approx 9\%$.

What causes this correction? As introduced in Sect. 1, a migrating $c_B$ peak is mirrored by a moving variation in porosity. This evolution implies freezing and melting on the peak's trailing edge and leading edge (Fig. 2c), causing water to be expelled from and absorbed into these regions, respectively, owing to the density change during phase change (see (19)). Hence a water flow localised about the peak transports solute in the same direction to speed up its migration. Although modest in size, the correction applies at all depths and also in the three-dimensional model.

## 2.5 Diffusion of impurity signals

The second (more crucial) departure from Rempel's predictions is that signals in $c_B$ will diffuse, as we emphasised at the outset. In Rempel's theory, $c_B$ signals suffer no distortion as they migrate under anomalous diffusion except for a slight amplitude change due to the right-hand side of (16). But as anticipated in Sect. 2.1, they will perturb $c(z)$ to drive solute diffusion (Fig. 2). The consequence can be studied via a stripped-down version of (15) (or (16)) where ice flow, water flow and vein motion (residual diffusion) are ignored:

$$\frac{\partial c_B}{\partial t}+\nabla.(\mathbf{u}_c c_B)=0 \ . \tag{21}$$

For the anomalous velocity, substituting for $c$ from (8) into (14) yields, after some algebra,

$$\mathbf{u}_c = -D\frac{\nabla c}{c} \approx \frac{D}{(T_0-T)}\left[\nabla T - \frac{\gamma T_0}{2\rho_i L}\sqrt{\frac{3\alpha(T_0-T)}{d_g^2\Gamma}}c_B^{-3/2}\nabla c_B\right]. \tag{22}$$

Note that this result assumes that $d_g$ varies slowly. If grain size fluctuates strongly over short distances (e.g. decimetres), the evaluation of $\nabla c$ would introduce another term involving $\nabla d_g$ (Sect. 3.4). With the present result in (22), (21) becomes

$$\frac{\partial c_B}{\partial t} + \nabla\cdot\left(\frac{D\nabla T}{T_0-T}c_B\right) = D\frac{\gamma T_0}{2\rho_i L}\sqrt{\frac{3\alpha}{d_g^2\Gamma}}\nabla\cdot\left(\frac{c_B^{-1/2}\nabla c_B}{\sqrt{T_0-T}}\right). \tag{23}$$

This partial differential equation implies not only signal migration driven by $\nabla T$, but also signal diffusion, due to the right-hand (Gibbs–Thomson) term, which encapsulates the $c$-perturbation. The diffusion is nonlinear and independent of $\nabla T$. Both
the diffusion and advection terms in (23) are controlled by $D$ because they originate from the molecular diffusion of solute in vein water. Setting $\gamma \to 0$ (which forces $\Gamma c \equiv T_0 - T$) recovers Rempel's theory.

    How fast do signals diffuse? One way to gauge their rate of broadening and damping versus migration is by taking the magnitude ratio of the two terms in the square bracket in (22):

$$\chi = \frac{\gamma T_0}{2\rho_i L}\sqrt{\frac{3\alpha(T_0-T)}{d_g^2\Gamma}}c_B^{-3/2}\frac{|\nabla c_B|}{|\nabla T|}. \tag{24}$$

The dimensionless number $\chi$, which also sizes diffusion against advection in (23), resembles the inverse of the Péclet number in fluid mechanics. But the context here is unique as our model concerns the transport of bulk concentration signals, and the underlying physics involve the geometry of the vein network and two-phase (solute–ice) thermodynamic equilibrium.

    As an example, taking $T_0 - T \sim 25$ K, $|\nabla T| \sim 20$ K/km $= 0.02$ K m$^{-1}$, $d_g \sim 5$ mm, $c_B \sim 1$ $\mu$M and $|\nabla c_B| \sim 1$ $\mu$M/10 cm $= 10^{-5}$ M m$^{-1}$ (a decimetre-scale signal), which roughly approximate conditions in the lower third of the GRIP and EPICA ice
cores, gives $\chi \sim 1.7$. The effective diffusivity in the right-hand term of (23) is then $2.1 \times 10^{-6}$ m$^2$ yr$^{-1}$, which is much higher than the diffusivities reported by Barnes et al. (2003) (Sect. 1). Although these ballpark estimates depend on the chosen values (including the signal magnitude) and will change if we consider elsewhere in the ice column, they show that diffusion can pervasively modify signals, whose short length scales play a key role in amplifying the perturbations on $c$ to cause diffusion at a rate rivalling migration. We confirm this by numerical simulation next.

## 3 Ice-core numerical experiments

Using the model above, which revises and extends Rempel's theory, we proceed to simulate the distortion of impurity signals in ice cores in both Greenlandic and Antarctic settings, by doping shallow ice with chemical peaks and seeing how they evolve. Most experiments explore predictions with a fully-connected vein network, but we include some that simulate partial disconnections ("blocked veins") by decreasing the molecular diffusivity $D$.

### 3.1 Material reference frame and set-up

A general simulation of (4), (5) and (17)–(19) would couple them to time-varying velocity, temperature and grain-size distributions in the ice column. But while the corresponding ice flow and thermal calculations are well established (e.g. Cuffey and Paterson, 2010), reliable grain-size modelling remains out of reach, especially for deep ice, where the way in which the mean grain size $d_g$ is governed by strain-induced recrystallisation processes is poorly understood (Faria et al., 2014; Ng and Jacka, 2014). Consequently, we prescribe time-invariant "background" profiles of $w(z)$, $T(z)$ and $d_g(z)$ in most of our experiments. In our final experiments in Sect. 3.4, we impose short-scale fluctuations on $d_g$ to see what could happen to the impurity evolution, in a simple way without modelling recrystallisation processes.

We track signals in a reference frame moving with the ice, because their length scale ($\lambda \lesssim 10^{-1}$ m) is much shorter than the core (ice thickness $H \sim 10^3$ m). The computational burden of modelling a short ice section rather than the whole core is also substantially less. To measure the separation distance of signals from ice material that was deposited on the surface at $t = 0$ and now lies at depth at time $t$ (so $t$ represents age), we use the displacement variable $z' = z - g(t)$, where $g(t)$ is the core's depth-age scale, defined by

$$t = g^{-1}(\zeta) = \int_0^\zeta \frac{dz}{w(z)} \, . \tag{25}$$

The change of variable from $z$ to $z'$ involves $\partial/\partial z \to \partial/\partial z'$ and $\partial/\partial t \to \partial/\partial t - g'(t)\partial/\partial z'$, with $g'(t) \equiv w(g(t))$, so (17) and (18) become

$$\frac{\partial \phi}{\partial t} + \tilde{w}\frac{\partial \phi}{\partial z'} = \frac{m}{\rho_i} + \frac{\partial}{\partial z'}\left(\kappa \frac{\partial \phi}{\partial z'}\right) \, , \tag{26}$$

$$\frac{\partial c_B}{\partial t} + \tilde{w}\frac{\partial c_B}{\partial z'} + \frac{\partial}{\partial z'}(qc) = \frac{\partial}{\partial z'}\left(\kappa \frac{\partial c_B}{\partial z'}\right) - c_B \frac{\partial w_c}{\partial z'} \, , \tag{27}$$

with the ice velocity in the new reference frame $\tilde{w}$ given by

$$\tilde{w}(z',t) = w(g(t) + z') - w(g(t)) \, . \tag{28}$$

This is the velocity field at time $t$ seen from the (moving) ice material at depth $g(t)$. Figure 3 depicts characteristic curves representing material trajectories on the $z$–$t$ and $z'$–$t$ plots under the typical compressive flow at a divide, where $-dw/dz$ is the

vertical strain rate. As a signal descends, layer-thinning causes it to narrow, but this is countered by our newly-discovered Gibbs–Thomson diffusion and residual diffusion, while anomalous (Rempel) diffusion displaces it from the ice if the temperature gradient is non-zero.

To derive a self-contained evolution equation for $c_B$, we combine (27), (26) and (19) by using the same substitutions as those leading to (20), finding

$$r\left(\frac{\partial c_B}{\partial t}+\tilde{w}\frac{\partial c_B}{\partial z'}\right)+w_c\frac{\partial c_B}{\partial z'}=-c_B\frac{\partial w_c}{\partial z'}+\frac{\partial}{\partial z'}\left(\kappa\frac{\partial c_B}{\partial z'}\right)-q\frac{\partial c}{\partial z'}-(1-r)\left[\phi\left(\frac{\partial c}{\partial t}+\tilde{w}\frac{\partial c}{\partial z'}\right)+c\frac{\partial}{\partial z'}\left(\kappa\frac{\partial \phi}{\partial z'}\right)\right]. \qquad (29)$$

The terms on the right-hand side here involving $c$ and $\phi$ are negligible, as scaling shows that they are $\sim \lambda(1-r)/H \sim 10^{-5}$ times of those terms on the left. This is verified numerically in all of our experiments. Therefore we approximate (29) as

$$r\frac{\partial c_B}{\partial t}+(r\tilde{w}+w_c)\frac{\partial c_B}{\partial z'}=-c_B\frac{\partial w_c}{\partial z'}+\frac{\partial}{\partial z'}\left(\kappa\frac{\partial c_B}{\partial z'}\right). \qquad (30)$$

In our simulations, (30) is solved by the explicit finite-difference method, with the anomalous velocity computed from $w_c = -(D/c)\partial c/\partial z$ (Appendix A details the solution for $c$ from (4) and (5)), the diffusivity $\kappa$ calculated from (9), and $\partial c_B/\partial z = 0$ prescribed at the $z'$-domain boundaries (this leads to no evolution there), far from the signal of interest. The doped signal is introduced at a depth near the firn-ice transition ($\approx 100$ m), in ice whose age $t$ corresponds to that depth. The subsidiary variables $\phi$, $r_v$, $m$ and $q$ are also calculated from $c$ and $c_B$ at each time step. Rempel et al. (2001) argued that $\left|c_B(\partial w_c/\partial z)\right| \ll \left|w_c(\partial c_B/\partial z)\right|$, but we do not ignore the term $-c_B(\partial w_c/\partial z')$ on the right-hand side of (30), because the full flux divergence $\partial(w_c c_B)/\partial z'$ is needed for solute conservation, i.e., no leakage.

We experiment with two sets of background profiles (Fig. 4), based on the glaciological conditions at the GRIP ice core site in central Greenland and the EPICA Dome C core site in Antarctica. In the GRIP runs, we use the depth-age scale $t(z)$ and velocity $w(z)$ from a Dansgaard-Johnsen model with the ice thickness $H = 3029$ m, the kink at 1000 m above the bed, and surface accumulation rate $a = 0.23$ m yr$^{-1}$ ice equivalent. In the EPICA runs, we use $t(z)$ and $w(z)$ from the model $w = m_{base} + (a - m_{base})[(H - z)/H)]^n$ (Ritz, 1992) with $H = 3275$ m, $n = 1.7$, $a = 0.023$ m a$^{-1}$ and the basal melt rate $m_{base} = 0.0008$ m yr$^{-1}$, which yields a depth-age scale approximating the one published by Parrenin et al. (2007). Smoothed versions of $T(z)$ and $d_g(z)$ measured at the ice-core sites are used (Fig. 4c–d, 4g–h). The prescribed profiles are exemplative only. In reality, ice at different depths has experienced different glaciological conditions due to changing accumulation, ice-sheet elevation and climatic temperature over interglacial–glacial time scales. Our interest is not in reconstructing the histories of these conditions.

## 3.2 Results: single-peak experiments

Figure 5a presents snapshots from a GRIP run of the evolution of a decimetre-scale signal doped as a Gaussian peak (grey curve: $c_B = 1 + 5 \exp[-(z'/\Delta)^2]$, with $\Delta = 0.08$ m) in ice 500 years old ($z = 112.4$ m). Initially the peak, centred at $z' = 0$, has a "full width at half maximum" (FWHM) of 0.13 m. Its set amplitude, 5 $\mu$M, is based on the size of commonly observed peaks

in ice-core records (e.g. $\sim 600$ $\mu$g L$^{-1}$ for SO$_4^{2-}$; $\sim 150$ $\mu$g L$^{-1}$ for Cl$^-$; $\sim 80$ $\mu$g L$^{-1}$ for Na$^+$). The peak decays rapidly in the first 20 kyr (upper 2 km at the GRIP site) with negligible migration, and migrates into $z' > 0$ more noticeably afterwards, as the ice section descends deeper where the temperature gradient increases (Fig. 4c). Movie S1 shows the full evolution of this control run. Strong diffusion of the signal is evident not just from the peak's decay, but also its broadening, which overcomes the effect of vertical compression. Recall that in the material reference frame, compression shortens the section continually,

so ice enters the simulation domain at both ends. Figs. 5b and c exemplify the perturbation on $c$ (caused by the $c_B$ peak) and the resulting large wiggle on the velocity $w_c$, which represents the $\gamma$-contribution in (22) and is what causes the Gibbs–Thomson diffusion. As in Rempel's theory, the signals on $\phi$ and $r_v$ are collocated with the $c_B$ peak throughout the evolution.

To check this diagnosis for the origin of signal diffusion, another run is conducted (Fig. 5d–f; Movie S1) with everything unchanged except that the Gibbs–Thomson term in (5) is turned off by setting $\gamma = 0$. As expected, the strong diffusion in the

control run disappears, as no perturbation now arises on $c$ and $w_c$, but there is still residual diffusion from vein motion. The peak narrows under vertical compression without much amplitude reduction until $t \approx 20$ kyr. It subsequently decays because strong $c_B$ gradients on its steepening sides amplify the residual diffusion, despite $\kappa$ being small ($\sim 10^{-8}$ m$^2$ yr$^{-1}$; Fig. B2). The peak's migration trajectory in this run is identical to that in the control run because migration is independent of the peak shape and the diffusion mechanisms. By $t \approx 100$ kyr ($\approx 2800$ m depth) it has displaced from the ice by $\approx 0.6$ m. A further experiment

with $\kappa = \gamma = 0$ (not shown) reproduces the "Rempel limit" of a migrating peak with no diffusion, as far as its diminishing width can be resolved by our $z'$-grid spacing, 0.0025 m. This implies that the simulated signal behaviour in the experiments is not due to numerical diffusion in our finite-difference scheme. Finally, repeating the control run with $\kappa = 0$ modifies the results in Fig. 5a only slightly, confirming that residual diffusion becomes important only when a signal becomes very narrow.

Figure 6a–c and Movie S2 present the control run for EPICA, where ice 4 kyr old ($z = 89.9$ m) is doped with the same

peak. The simulated behaviour is similar to that in the GRIP run, but occurs on a much longer time scale due to the low accumulation rate at the EPICA site. The peak migrates from the start because a sizeable temperature gradient spans the ice column (Fig. 4g). Low compressive strain rate, coupled with slow ice submergence and long time for diffusion, yields a wider

peak at all depths than in the GRIP run that has a vastly increased "age span" (i.e., the peak's width in the age domain; discussed later in Fig. 8c, f) compared to the doped signal. Again, comparison against a run with $\gamma = 0$ (Fig. 6d–f; Movie S2) confirms

the Gibbs–Thomson perturbation as the cause of signal damping and broadening, and illustrates the weaker residual diffusion.

The rapidity of signal widening versus migration in distorting the peak in both control runs (Figs. 5a and 6a) is anticipated by the non-small dimensionless number $\chi$ in Sect. 2.5. According to (24), near-constant temperature in the top half of the GRIP column (Fig. 4c) preconditions a large $\chi$ there. Indeed, signal diffusion dominates that part of the GRIP control run, confirming also its independent operation from migration. Deeper in both cores, migration becomes more significant as $\chi$ is

reduced by higher $T$ and higher $dT/dz$. In theory, larger grain sizes near the bed (Fig. 4d, h) also slow the rate of the simulated signal decay, but the peaks in our control runs have long dissipated before reaching such depths.

These initial runs demonstrate the signal migration of Rempel's theory, but paradoxically highlight that signals may not survive deep into the ice where it predicts their displacement to become so large to be palaeoclimatically important. More precisely, some remnant signals always survive, but with such small amplitudes and such large age spans compared to the

original signals that all essential palaeoclimatic information has been lost. There is an apparent problem to resolve, as distinct deep ionic peaks are found in many ice cores (Fig. 1) (e.g. Röthlisberger et al., 2008; Traversi et al., 2009; Svensson et al., 2013; Tison et al., 2015; Schüpbach et al., 2018), although they may be due to impurities outside veins.

Sticking with the vein model for now, can peaks with a different shape survive damping and broadening to reach deep ice? We study this by changing the width of the doped peaks, as this alters their flank gradient, which is a key control of their

diffusion rate. Sensitivity experiments are conducted by varying the width parameter $\Delta$ of the Gaussian function between 0.02 and 0.32 m (with the control run parameters unchanged), and by tracking the amplitude, FWHM (full width at half maximum) and age span of the peak in each simulation. The age span is found by dividing the FWHM by the local ice velocity $w$.

Figures 7 and 8 plot – for GRIP and EPICA, respectively – the evolving peak morphometry in these "$\Delta$ experiments" (grey curves). The control runs (black) and the runs where the Gibbs–Thomson effect has been turned off (orange) are included

for comparison. As shown by the grey curves, doping a narrower initial peak hampers its survival, as its steep sides cause strong diffusional draw-down of amplitude; broader peaks retain amplitude for longer but meet the same fate as they narrow under vertical compression. We observe an interesting feedback between width and amplitude evolution. Compression steepens the flanks of signals to accelerate their damping, whereas amplitude reduction makes them shallower and less prone to damping and broadening. Thus the compressive strain rate is a key driver of signal diffusion. The balance of compression

and broadening causes different peaks to end on similar width trajectories at depth (panels b & e, Figs. 7 and 8). Accordingly, peaks with different initial time durations acquire near-equal age spans increasing down core (panels c & f, Figs. 7 and 8), which define the minimum time resolution for deep climate signals. These interactions are absent from the study of Rempel et al. (2001), who did not simulate signal shape evolution. Their companion papers (Rempel et al., 2002; Rempel and Wettlaufer, 2003) did so but excluded layer-thinning and diffusion.

So, can single peaks survive into deep ice? The Δ experiments show that peaks at decimetre/centimetre scale struggle to do so. Even for initially wide peaks (e.g. Δ = 0.32 m) near the firn-ice transition, the Gibbs–Thomson diffusion has reduced their amplitude four-fold by the time they reach $z \approx 2300$ m at GRIP (where the age is $\approx 25$ ka), and 2000 m at EPICA ($\approx$ 175 ka). Setting $\gamma = 0$ prolongs the signals' survival (Figs. 5–8), but residual diffusion still prevents them from reaching the lowest several hundred metres with a sizeable fraction of their original amplitude, not to mention that ignoring the Gibbs–

Thomson effect is unphysical.

In the present theoretical framework, is there any way for signals to reach deep ice without losing integrity (amplitude, narrowness)? One possibility is the suppression of solute transport by partial vein blockage/disconnection, which we simulate here in a crude manner by artificially decreasing the molecular diffusivity $D$ – this cannot capture heterogeneous vein transport at the grain scale. In Figs. 7 and 8, the blue curves plot the results of simulations with $D$ suppressed by different factors. The

same doped peak and parameters of the control runs are used otherwise. A suppression factor of 0.001–0.01 postpones signal decay to a similar extent as turning off the Gibbs–Thomson effect. The lesser factor (0.001) allows the peak to reach 2750 m with half its original amplitude. Even with such strong suppression, however, peak survival is hindered in deeper ice because the low strain rate there (Fig. 4b, f) provides ample time for signal diffusion to occur, and because rising temperature near the bed increases $\kappa$. Note that in Figs. 7 and 8, a perfectly-preserved peak signal that does not diffuse would have constant

amplitude and age span, and its FWHM would decrease towards the bed as a result of vertical compression.

The simulated displacement, age offset and age span of the peaks are of potential palaeoclimatological interest. Figure 9 shows that in the control runs the peaks displace by $\sim 1$ m or more in deep ice, causing their apparent age to exceed their true age by hundreds of years in ice $\approx 100$ ka at GRIP, and by several thousand years in ice $\approx 500$ ka at EPICA. Since the migration rate is independent of the signal shape, the Δ experiments yield the same displacements and offsets as the control experiments.

For both sets of experiments, Figures 7 and 8 (c & f) show that peaks arriving in deep ice have age spans of several hundred years at GRIP and several thousand years at EPICA (approaching the precession time scale in ice 700–800 kyr old); however, we caution against using these results to evaluate deep climatic histories retrieved from ice cores, because these sets of experiments predict near-zero signal amplitude at such depths. In contrast, decreasing $D$ suppresses both signal migration and diffusion (see (23)), so the corresponding peaks remain much narrower during their evolution (Figs. 7 and 8, b & e) and migrate

much less than in the control/Δ runs (Fig. 9, numbered curves). A suppression factor of 0.001–0.01 enables a peak with FWHM < 0.2 m, age span < 200 yr, and potentially detectable amplitude to reach $\approx 2900$ m depth, with an age offset of < 50 yr at GRIP and < 300 yr at EPICA. These numerical findings are illustrative, as they depend on the depth-age scale assumed for each site (notably its precise behaviour at depth) and are limited by the fact that we are not solving the inverse problem with time-varying palaeoclimatic forcing. Using them to interpret specific details of the ice-core records is not advisable at this

stage also because how much matrix/grain-boundary impurities contribute to those records is unknown (Sect. 4).

For completeness, all of the above experiments have been repeated with doped peaks with twice amplitude (10 $\mu$M), to cater for some especially high (rarer) peaks in the observed records, which may have more chance to survive. Although the corresponding remnant signals retain greater bulk concentrations at all depths than before, their pattern of decay relative to the initial amplitude and the FWHM and age span results are only marginally altered (Figs. S1 and S2 – see Supplement; cf. Figs. 7 and 8). Our single-peak experiments thus confirm the difficulty for palaeoclimate information to be preserved at depth.

### 3.3    Results: multiple-peak experiments

The diffusion of $c_B$ means that neighbouring peaks can merge as they descend the ice column. This process is illustrated in Figure 10 and Movie S3 by a simulation with two peaks. Their merging begins at $\approx$ 7 kyr; the deeper peak moves towards $z'$ = 0 due to vertical ice compression; a bimodal signal ceases at $\approx$ 12 kyr. Such merging suggests a second explanation for why distinct peaks can feature in deep ice even with strong damping: instead of deriving from a single peak high up in the column, a deep peak might form by the agglomeration of multiple signals/peaks as these merge under compression. This signal-forming mechanism may not be evident from the $c_B$ profile measured from ice cores, which gives an instantaneous record of the signals.

To test this idea, in the next experiments we simulate the evolution of multiple signals doped in shallow ice stretches 20 m long at GRIP and 80 m long at EPICA. Three runs are made for each site, one with the control-case parameters and two with $D$ suppressed by 0.1 and 0.03 (to simulate vein blockage), with both the Gibbs–Thomson effect and residual diffusion included, and using an initial $c_B$ profile formed by adding many Gaussian peaks (numbering 300 at GRIP and 1200 at EPICA) of random amplitudes, widths and positions onto a 1 $\mu$M base level (Fig. 11a). Movies S4 and S5 document these runs.

We focus our analysis on the EPICA runs (Fig. 11; Movie S5), as the GRIP findings are qualitatively similar (although things occur faster there). In the control run (black curves), strong damping and merging smooth the signals rapidly, so $c_B(z')$ retains long-scale variations only – and no peaks – at depth. This outcome is consistent with what we learned from the single-peak experiments. When $D$ is reduced (blue and red curves), compressional shortening, with the now slower diffusion, causes bundle of peaks to merge into new signals that subsume their solute content. This process operates continuously on all signals, with stretches having a high density of peaks turning into peaks, and stretches having a low density into troughs. The vertical compression is crucial in helping signals maintain their integrity against diffusion.

When $D$ is suppressed by 0.03 (Fig. 11, red curves), we see distinct peaks persisting in deep ice, many traceable back in time to predecessor groups of peaks, rather than a single peak (e.g. dashed boxes). The balance of diffusion and shortening here is such that the deep peaks have similar widths as their shallow counterparts ($\sim$ dm), despite an overall reduction of signal amplitude with depth. The ice in Fig. 11d has shortened by approximately ten times since the start of the run, so each peak there encapsulates the signals and solute of an original interval some ten times longer. Signal survival here is aided by the enhanced survival of single peaks due to decreased molecular diffusivity (Sect. 3.2), but also involves the lumping of solute

from neighbouring peaks. We find in further experiments (not shown) that when $D$ is reduced even more (suppression factor $\lesssim 0.001$), the peaks continue to narrow into the $\sim$ cm range at depth. Fig. 11 shows an effect already known from the single-peak experiments: a decrease in $D$ reduces signal displacement as well as broadening.

The foregoing experiments demonstrate how long-scale averages on $c_B$ at shallow depths – reflecting long-term background levels of impurity input at an ice-core site – evolve to become meaningful variations at depth, as signals are compressed and their fine details filtered out by diffusion. In Fig. 11, the mean level of the 3,600-year long signal sequence is preserved at depth as a bump (of the same duration) about 7 $\mu$M above the surrounding ice. Ice-core analyses of the major ions frequently interpret deep features of this kind as reflecting real palaeoclimatic variations on time scales of $10^1$–$10^2$ kyr (e.g. Mayewski et al., 1997, Wolff et al., 2006; Schüpbach et al., 2018); it is understood that fewer high-frequency palaeoclimatic details are retrievable from deeper ice, due to the finite resolution of ice-core sampling, alongside layer thinning, which causes more time to be encapsulated in a given ice thickness. Our simulations highlight the Gibbs–Thomson effect in vein impurity transport as a further cause of low-pass filtering. Moreover, they show that long-scale signals will migrate under Rempel's anomalous diffusion mechanism unless the vein network is partially disconnected (see the bump's locations in Fig. 11e).

### 3.4   Grain-size fluctuations and how they create $c_B$ signals

Our experiments have used smooth $d_g$ profiles so far, as the lack of a robust model of grain-size evolution precludes intimate study of how this process intercouples with vein impurity transport. Ice-core records exhibit grain-size fluctuations (e.g. Fig. 4d, h), although continuous measurements of $d_g$ at a high (e.g. decimetre) resolution over long core sections remain rare. How might rapid fluctuations in $d_g$ impact $c_B$? Here we study this topic with more analysis and a few tentative simulations, discovering a mechanism by which new impurity signals can form post-depositionally as a result of grain-size fluctuations.

In this connection, in the Interactive Discussions of our manuscript (see interactive comments on The Cryosphere Discussions, RC2 (https://doi.org/10.5194/tc-2020-217-RC2, 2020) and RC3 (https://doi.org/10.5194/tc-2020-217-RC3, 2020) and associated author comments AC2, AC3 and AC4) it was clarified that Rempel et al. (2001) neglected the Gibbs–Thomson effect from the liquidus relation by making the assumption that the vein radii $r_v$ were spatially uniform – the justification for this being an anticorrelation between mean grain size and impurity loading, which has been observed in ice-core records (see Thorsteinsson et al. (1995), Alley and Woods (1996) and Thorsteinsson et al. (1997) for evidence related to soluble ions). The concept is that grain recrystallisation processes cause $d_g$ to respond to $c_B$ signals at decimetre/centimetre scale to prevent variations in $r_v$ at that scale. Thus, whereas Rempel et al. (2002) and Rempel and Wettlaufer (2003) ignored the interfacial effect by invoking *smallness* of the Gibbs–Thomson term, the alternative assumption addressed here invokes its *constancy*. Equations (7) and (8) in our model and their general form based on (5) show that this assumption holds if $d_g^2 c_B =$ constant. In the Interactive Discussions (see interactive comment on The Cryosphere Discussions, RC2, https://doi.org/10.5194/tc-2020-217-RC2, 2020), it was suggested that the processes in glacier ice might ensure this inverse-

square coupling ($d_g \propto c_{\mathrm{B}}^{-1/2}$) through the ice column, so that signals migrate by anomalous diffusion, as conceived in Rempel's theory, without the strong (Gibbs–Thomson) diffusion found in our study.

While such theory is difficult to prove or disprove until firm quantitative modelling has been offered to explain the causal mechanisms in ice linking grain size to impurities, some obstacles for it are apparent: (i) For diffusion to vanish, $d_g{}^2 c_{\mathrm{B}} =$ constant needs to be obeyed identically. Anticorrelation between $d_g$ and $c_{\mathrm{B}}$ does not generally imply suppression of diffusion, since the analysis of Sect. 2.5, repeated with $d_g \propto c_{\mathrm{B}}^{-\beta}$ ($\beta > 0, \neq 1/2$), shows that a diffusion term still arises in (23), just with a different form (and with either higher or lower rate) than before. (ii) Whether $d_g$ and $c_{\mathrm{B}}$ are anticorrelated in ice cores is in fact unknown, because the observed anticorrelation used to support the theory concerns their total impurity loading, which includes an unknown contribution of matrix and grain-boundary impurities: it is the vein impurity component, not the total, that must satisfy $d_g{}^2 c_{\mathrm{B}} =$ constant. (iii) Existing glaciological theories on how soluble ions affect the mean grain size consider the effect of such impurities at grain boundaries, through drag production to limit grain-boundary mobility (e.g. Alley et al., 1986a, 1986b), not the effect of impurities in the veins. Also, as $r_v$ is uniform in the suggested theory, the solute concentration of the vein water ($c$) will be uniform even where $c_{\mathrm{B}}$ has a signal (see (7)); hence it is elusive how $c_{\mathrm{B}}$ can influence grain-boundary motion to control $d_g$. It is also unclear how $d_g{}^2 c_{\mathrm{B}} =$ constant is to be maintained despite other grain-size controls, e.g., microparticle abundance, stored strain energy (Faria et al., 2014; Ng and Jacka, 2014). A further complexity that hampers such maintenance in glacier ice is that the melting point depression is regulated by the overall mix of multiple species of dissolved chemicals, while individual species influence the grain size differently. In the light of these considerations, the idea of signal migration with diffusion eliminated by a highly-specific coupling between $d_g$ and $c_{\mathrm{B}}$ seems problematic.

The general question about the impact of grain-size fluctuations on $c_{\mathrm{B}}$ is still of interest. Partial insights into this can be gained with our model. In the diffusion analysis of Sect. 2.5, if $d_g$ varies on short distances, then the anomalous velocity found by substituting for $c$ from (8) into (14) (using the chain rule when differentiating $d_g^{-1} c_{\mathrm{B}}^{-1/2}$) has a longer form

$$\mathbf{u}_{\mathrm{c}} = \frac{D}{(T_0 - T)} \left[ \boldsymbol{\nabla} T - \frac{\gamma T_0}{2\rho_i L d_g} \sqrt{\frac{3\alpha(T_0 - T)}{\Gamma}} c_{\mathrm{B}}^{-3/2} \boldsymbol{\nabla} c_{\mathrm{B}} - \frac{\gamma T_0}{\rho_i L d_g{}^2} \sqrt{\frac{3\alpha(T_0 - T)}{\Gamma}} c_{\mathrm{B}}^{-1/2} \boldsymbol{\nabla} d_g \right] \tag{31}$$

(cf. (22)), and the signal evolution equation in (21) becomes

$$\frac{\partial c_{\mathrm{B}}}{\partial t} + \boldsymbol{\nabla} \cdot \left( \frac{D \boldsymbol{\nabla} T}{T_0 - T} c_{\mathrm{B}} \right) = D \frac{\gamma T_0}{2\rho_i L} \sqrt{\frac{3\alpha}{\Gamma}} \boldsymbol{\nabla} \cdot \left( \frac{c_{\mathrm{B}}^{-1/2} \boldsymbol{\nabla} c_{\mathrm{B}}}{d_g \sqrt{T_0 - T}} \right) + D \frac{\gamma T_0}{\rho_i L} \sqrt{\frac{3\alpha}{\Gamma}} \boldsymbol{\nabla} \cdot \left( \frac{c_{\mathrm{B}}^{1/2} \boldsymbol{\nabla} d_g}{d_g{}^2 \sqrt{T_0 - T}} \right) . \tag{32}$$

As in Sect. 2.5, this model equation treats the scenario without ice deformation/compression. The extra term involving $\boldsymbol{\nabla} d_g$ on its right-hand side (cf. (23)) indicates that grain-size fluctuations will generate signals even if $c_{\mathrm{B}}$ has no initial signals. This source term balances with the diffusion and advection terms at large $t$ to cause a steady-state signal locked to the $d_g$ fluctuation. The term representing Gibbs–Thomson diffusion still operates, although it does not damp the signal completely when a source term is present. If $\boldsymbol{\nabla} T = 0$, steady state is achieved with $d_g{}^2 c_{\mathrm{B}} =$ constant (so $c_{\mathrm{B}}$ mirrors the $d_g$ fluctuation). This is the outcome

of the evolution, not the external process coupling that is prerequisite for signal migration without diffusion (discussed above), because $\nabla T = 0$ implies zero advection. If $\nabla T \neq 0$, the steady-state signal will not satisfy $d_g^2 c_B$ = constant.

The mechanism creating a new signal can be explained in terms of vein interactions, most easily by considering ice with uniform $c_B$ initially, without compression or temperature gradient. A short stretch with smaller grain size has higher vein length
density and lower $r_v$ (see (1) and (4)). This implies a stronger Gibbs–Thomson effect (in (6)), so that, at a given temperature, the melting-point depression due to solute must be less, i.e., $c$ must be lower in the stretch than in the surrounding ice. The smaller grain size thus sets up a negative perturbation in $c$ – by lowering the curve of equation (4) and the intersection point in Fig. 2f – and the resulting concentration gradients drive solute diffusion that raises $c_B$ over the stretch. In this example, the $c_B$ signal will grow until $r_v$ becomes spatially uniform. This happens when $d_g^2 c_B$ attains the same value everywhere; the vein
conditions inside and outside the stretch are then characterised by the same intersection point in Fig. 2f. If the ice deforms or a temperature gradient exists, a signal would still emerge but its evolution and final shape (if steady) will be different.

To verify these expectations, we perform modified GRIP and EPICA control runs, using the same simulation model and background profiles as in Sect. 3.1, without doping an initial signal on $c_B$. For $d_g$, we impose a constant background of 4 mm for simplicity, with a fluctuation $\Delta d_g = \pm 3 \exp[-(z'/0.1)^2]$ superposed on top (Fig. 12a, c, e). Simulations are made with $\Delta d_g$
negative or positive – representing, respectively, grain fining or coarsening in a decimetre-wide band – and with $\Delta d_g$ having a fixed width or narrowing under compression (the $c_B$ signal is always vertically compressed). Thus four runs are made for each site. They are artificial as our $d_g$ prescriptions are not physically based or necessarily realistic.

Figure 12 and Movie S6 show the GRIP results. In the two runs with negative $\Delta d_g$ (Fig. 12a–d), a W-shaped signal in $c_B$ emerges in the first few kyr and grows more slowly to a height of $\sim 10\ \mu$M. Its form evolves over the kilometre depth scale
and does not reach steady state due to continued compression and temperature changes at depth ($r_v$ does not become uniform in long time because of these and $\nabla T \neq 0$; Movie S6). Its peak is narrower than the $d_g$ fluctuation at all times. Compression of the grain-size fluctuation creates a narrower signal and accelerates signal growth (Fig. 12c, d) by causing high spatial gradients in $d_g$ (see the source term in (32)). In the two runs with positive $\Delta d_g$, a much smaller S-shaped signal of $\sim 1\ \mu$M forms (Fig. 12e, f; Movie S6). The EPICA runs show similar evolution over a longer time (Fig. S3; Movie S7). Note that the Rempel et
al. (2001) equations cannot uncover these findings because they do not contain grain size as a variable, and all signals simulated here are localised by the grain-size fluctuation and do not migrate relative to the ice, despite non-zero advection. Furthermore, (32) indicates that vein blockage/disconnection (suppression of $D$) would slow the growth of signals.

In the mechanism found here, grain fining creates large peaks in $c_B$, whereas grain coarsening creates small signals, so an anticorrelation between $c_B$ and $d_g$ is expected for signals with this origin. This property – demonstrated for signals that do
not migrate by anomalous diffusion – gives another reason why anticorrelation between $c_B$ and $d_g$ does not guarantee signal migration without damping. The $c_B$ signals here can begin to form at any depth, wherever recrystallisation (e.g. strain-induced)

processes perturb $d_g$ at a short scale; signals where $d_g$ varies slowly still decay by Gibbs–Thomson diffusion. The associated concern that ice-core records may contain some signals unrelated to palaeoclimatic events is considered in the next section.

## 4   Discussion and conclusions

For two decades, Rempel's theory has raised concerns that palaeoclimatic signals in the soluble ion records of ice cores may have displaced by anomalous diffusion and suffer large age discrepancies, especially in the older, deeper parts of the records. Objections to signal migration invoke impeded or insignificant solute transport through veins – that the veins are partially disconnected (blocked by solid impurities and bubbles) or that most chemical signals reside outside veins, in the ice matrix or at grain boundaries.

A more fundamental issue with Rempel's theory is explained herein. While signals on $c_B$ (the bulk concentration of an ion species in the veins) unrelated to grain-size fluctuations can migrate in a connected vein system, a strong Gibbs–Thomson diffusion damps them, preventing those signals at decimetre or shorter scale from penetrating into deep ice. Only much longer background variations in $c_B$ can survive the diffusion to exhibit migration, and signals created by grain-size fluctuations do not migrate (not unless these move relative to the ice by extraneous mechanisms). *As the physics predicts that no/few migrating*
*short signals survive into deep ice where their displacement matters, the original concerns are no longer valid.* In our revised theory, signal damping is aided by a weaker residual diffusion due to stochastic vein motion. Modifying the derivation of this residual diffusion (Appendix B) to include accelerated grain-boundary motion during migration recrystallisation (Duval and Castelnau, 1995) or grain-growth driven diffusion (Barnes et al., 2003) strengthens our primary conclusion.

The conclusion is unaffected if we consider multiple solute species interacting via the liquidus (Rempel et al., 2002;
Rempel and Wettlaufer, 2003). These authors showed that the $c_B$ signals of different species would line up as they evolve, with periodic signals becoming in phase, and peaks in each species inducing collocated "sympathetic peaks" in other species; these adjustments occur in a time of $\lambda^2/D \sim 1$–10 yr for short signals ($\lambda \sim 10^{-1}$ m). Extending our theory for multiple species would thus add to the outcome an initial fast alignment of signals, before they evolve by the mechanisms studied herein.

What do our findings mean for the integrity and interpretation of ion records from ice cores? What explains the occurrence
of well-defined signal peaks deep in those records? To ponder these, it is useful to start with two end-member scenarios:

*Scenario 1: Vein-dissolved ionic impurities* ($c_B$) *comprise the main contribution to an observed ice-core record, with matrix/grain-boundary impurity contributions negligible.* In this scenario, observed deep peaks can only be explained by (i) widespread vein disconnection (which we modelled by reducing the molecular diffusivity $D$; Sect. 3.2) and/or (ii) signal formation induced by grain-size fluctuations (Sect. 3.4). The GRIP and EPICA simulations show that in (i),
the suppression factor on $D$ needs to be $\lesssim 0.001$ for shallow single peaks to survive into depth, but weaker suppression ($\sim 0.01$) allows some deep peaks to persist via compression-diffusion merging of signals. Figs. 7–9 suggest the

possibility of determining the suppression factor from the width, amplitude and position of the deep peaks, and then finding their age offset. In practice, the unknown initial peak size/shape will introduce uncertainty to this estimation.

This scenario spells good news for a key aspect of ice-core palaeoclimatic interpretation. Because reduced vein transport limits signal migration (Fig. 9) and because signals created by grain-size fluctuations are "localised" by them (Fig. 12), the deep peaks would not have displaced far – or at all – to accrue large age offsets. However, diffusional merging means that some signals may be a distorted, lumped signature of multiple climate events. The Gibbs–Thomson diffusion can smooth details more than the low-pass filtering caused by finite resolution of the impurity measurements – typically, $\sim 10$ cm in traditional ice-core sampling, and $\approx 1$ cm or less if using Continuous Flow

Analysis (Kaufmann et al., 2008; Bigler et al., 2011). At the model ice-core sites, the highest time resolution of climatic information retrievable from $c_B$ is quantified approximately by the depth-dependent age spans in Figs. 7 and 8.

        Our modelling reveals a new issue if Scenario 1 applies: some observed peaks may have formed as a result of grain-size fluctuations (on short length scales) instigated by glaciological processes during or after deposition. This process, which could impact multiple ion species at the same positions down-core, arises from the effect of grain size

on vein equilibrium thermodynamics. It may be misleading to read such signals for palaeoclimate information in the usual way for a given ion, although the grain-size fluctuations producing the signals may have a palaeoclimatic origin.

*Scenario 2: Impurities in the ice matrix (e.g. in salt micro-inclusions) and/or at grain boundaries dominate the ice-core record*, and their relative immobility explains the presence at all depths of prominent peaks, which experience vertical compression but do not diffuse or migrate by our mechanisms (they may be modified by slow diffusion

through ice or along grain boundaries). In this scenario, signals on the minor vein component ($c_B$) still evolve if the veins are connected – where grain size does not fluctuate rapidly – to migrate, decay, broaden and merge into long-scale variations.

The reality may be a mixture of these scenarios, with vein and matrix/grain-boundary impurities responsible for different signals on an ion record; their contributions may vary down-core, and between cores. Thus generally a record may be the sum

of an evolved (e.g. diffused, migrated) component and a largely unmodified component. Shallow signals could source from both components, as firn metamorphism apportions impurities to crystal grains and the premelted liquid. Matrix impurities may dominate deeper, as vein signals decay where the grain size varies slowly. If grain growth and recrystallisation relocates some matrix impurities to grain boundaries and hence to the veins, the signals in $c_B$ could be continually refreshed (at any depth). Such impurity transfer has been suggested (Glen et al., 1977; Mulvaney et al., 1988; Rempel et al., 2001), but also

debated (Ohno et al., 2005; Eichler et al., 2019), while it is understood that the apportioning and transfer depend on the ion species (Wolff et al., 1988). The creation of new signals by grain-size fluctuations complicates the vein impurity contribution, as part of the signal evolution becomes coupled to recrystallisation processes that determine historical changes of the ice texture. These considerations caution against interpreting all observed ionic signals directly for palaeoclimatic events and

variations: some signals may be distorted in form and duration, and some peaks may be caused by local grain fining (induced by recrystallisation processes (Faria et al., 2014) or high levels of dust/microparticles in the ice (Alley et al., 1986a)). Disentangling the vein and matrix/grain-boundary impurity components of a given ion record and their post-depositional evolution histories, and comparing the record with grain-size profiles to discern peaks with a potential "grain-size induced" origin, may be necessary for robust reconstruction. Note that our ideas do not oppose the view that most/much ionic impurity occurs in the matrix (Ohno et al., 2005), which does not strictly rule out the presence of any vein impurities.

Based on these considerations, we conclude that distinct deep peaks seen on a record may indicate (i) matrix impurities dominating the record, or (ii) relevance of both matrix and vein impurity signals, with the latter damped out at depth or preserved by vein disconnection, or (iii) a dominance of vein impurities in a disconnected vein network, or (iv) a dominance of vein impurities in connected veins that receive recent/sporadic impurity input from the matrix and/or where grain-size fluctuations create and maintain impurity peaks. In each case, the observed peaks would not have migrated or migrated far, although some diffusional merging of signals may occur. Unless many peaks owe their origin to grain-size fluctuations, the limited distortion of peaks inferred for all four cases here is consistent with the signal replicability observed between nearby ice cores (Wolff et al., 2005; Gautier et al., 2016), and supports the use of major ion records for synchronising ice-core age scales. We expect the glaciological conditions at different ice-core sites to cause contrasting distortion of signals and different patterns of grain-size induced peaks. Even if two sites receive identical peak signals at the firn-ice transition (there are no differences in atmospheric transport and deposition of the species and in its modification in firn), variations between their records – for the same ion – will result from differences in strain rate, ice temperature, and dust content/bubble density (which affects the level of vein blockage and recrystallisation processes). Some peaks in one record may be absent or more damped in the other; a group of peaks in one record may appear as a single merged peak in the other record. These variations, which are well known in ice-core studies, impact the identification and matching of peaks and peak sequences. The potential corruption of some records by abundant "grain-size induced" signals is an emergent problem that should be studied further.

As we have modelled vein impurity transport only, it is beyond our scope in this paper to disentangle the vein, matrix and grain-boundary components of any specific (observed) ion records. In fact, some ionic impurities in the veins may not be dissolved. An example is NaCl, whose eutectic point at $\approx$ –23 °C (pressure-dependent) suggests that it could precipitate in cold ice near the surface but re-dissolve in vein water in warmer ice at depth. Also, our analysis is not aimed at resolving where ionic impurities reside in glacier ice. Recent investigations of ice-core samples using Raman spectroscopy (Barletta et al., 2012; Eichler et al., 2019) have yielded a varied picture regarding the distribution of vein versus matrix impurities in ice, with the former authors finding abundant sulphate and nitrate, but the latter inferring a lack of ionic impurities, at triple junctions (i.e. veins). With an extended literature reporting different results on the subject (see Barnes and Wolff (2004) and the review parts of the two papers cited above), this state of knowledge suggests that one should not generalise any particular impurity distribution to all ice. Striving to understand the range of processes of impurity movement and segregation and their controls, and how they can cause different impurity distributions, is more important.

Hopefully, with better understanding in this direction, research will be able to develop the theory further by coupling our solute and porosity evolution equations for the veins with equations for the formation, transport and modification of matrix impurity sites – going beyond a static partitioning of the vein and matrix impurities (Rempel et al., 2002; Rempel and Wettlaufer, 2003). Another direction alluded to before is coupling with grain-size evolution. An extended theory could help palaeoclimatic studies more directly, in terms of quantifying post-depositional changes of measured records, revealing their artefacts, and developing refined palaeoclimatic inversions. We know too little at this stage to envision the details, but such theory will need to address a multi-directional transfer of ionic impurities between matrix, grain boundaries and veins. Other foreseeable complications include chemical reactions, grain-boundary motion influenced by impurities, and precipitation and dissolution of different impurities at different temperatures. Work that aids the theory development includes (i) a systematic study of solute signals in ice cores that quantifies their depth-varying spectral content, peak density and peak size-/width-frequency statistics and compares the signals to high-resolution grain-size measurements, and (ii) controlled laboratory experiments on ice samples to recreate evolving signals for testing the theory.

Finally, our model may be used to quantify the degradation of vein ionic signals during ice-core storage. The rates of residual diffusion, Gibbs–Thomson diffusion and signal migration (if storage imparts non-zero temperature gradient through a sample) are all minimised at low temperatures (Fig. B2 and (23)). Model runs (Movie S8) show that in ice with a mean grain size of 5 mm and containing a 10 $\mu$M high, $\approx$ 10 cm wide signal in $c_B$, the total diffusion in 100 years would reduce the signal amplitude by only 5% and 16% (with negligible broadening) if the sample is stored at –15 °C and –5 °C, respectively. These upper-bound reductions assume no vein blockage, and the signal diffusion time scale is still several kyr (Movie S8).

## Appendix A: solving for vein conditions

From Dani et al. (2012), relevant empirical formulas for the temperature depressions $\theta_c$ and $\theta_p$ in (5) (in Kelvin) are

$$\theta_c = k_1 c + k_2 c^2 + k_3 c^3 \quad \text{and} \quad \theta_p = a_1 p + a_2 p^2, \tag{A1}$$

where $c$ is measured in M, pressure $p$ is in Pa, and the constants are $k_1 = 4.7971$, $k_2 = -1.188$, $k_3 = 0.685$, $a_1 = 7.61 \times 10^{-8}$, $a_2 = 1.32 \times 10^{-16}$. At a depth $z$, we calculate the overburden pressure as $p = \rho_i g z$ without correcting for firn density. Then combining (4) and (5) yields

$$f = (T_0 - T - \theta_p) - k_1 c - k_2 c^2 - k_3 c^3 - \frac{\gamma T_0}{\rho_i L} \sqrt{\frac{3\alpha c}{c_B d_g^2}} = 0. \tag{A2}$$

This equation has one positive real root for $c$. We calculate it numerically with Newton's method, choosing $c = (T_0 - T - \theta_p) / k_1$ as the initial guess.

**Appendix B: residual diffusion of $c_B$ and $\phi$**

Consider, in polycrystalline ice, a three-dimensional network of veins with random orientations, which, as the grain boundaries migrate, move in random directions (Fig. B1). Here we show that this can cause diffusion of the bulk solute concentration $c_B$ and porosity $\phi$. For simplicity, we assume the vein motion to be isotropic.

As each vein segment migrates, its motion takes along pore space and solute (we ignore vein water flow, ice deformation, and other processes considered in the main model; Sect. 2.2). Transport arises from vein segments moving in myriad directions. Segments with the same size and solute content moving in opposite directions cancel in terms of contribution. A statistical description is needed to calculate the net effect. Suppose their migration velocities $\mathbf{v} = (v_x, v_y, v_z)$ across a given plane (Fig. B1) follow the probability density function $f$, so that the proportion of vein segments with velocity near $\mathbf{v}$ (in an incremental box $dv_x$–$dv_y$–$dv_z$) is

$$f(v_x, v_y, v_z)dv_x dv_y dv_z = F(v)v^2 \sin\theta' \, d\theta' d\phi' dv \,. \tag{B1}$$

The right-hand side puts $d\mathbf{v}$ in spherical coordinates (we dashed the symbols of the polar angle $\theta'$ and azimuth $\phi'$ to distinguish them from $\phi$ and $\theta$). Under isotropic migration, $F$ is a function of speed $v = |\mathbf{v}|$, independent of direction, and one may suppose $F$ decays to 0 as $v \to \infty$. This formulation resembles the kinetic theory of gases, where $F$ is the Maxwell-Boltzmann distribution (Chapman and Cowling, 1953).

Regardless of the exact form of $F$ for veins, their mean migration speed is

$$\bar{v} = \int_0^\infty \int_0^{2\pi} \int_0^\pi F(v)v^3 \sin\theta' \, d\theta' d\phi' dv = 4\pi \int_0^\infty v^3 F(v) \, dv \,. \tag{B2}$$

We expect $\bar{v}$ to be similar to the speed of grain boundaries. An estimate for the latter can be found from the normal grain growth law (Gow, 1969; Duval, 1985; Cuffey and Paterson, 2010),

$$\frac{d(d_g{}^2)}{dt} = K \,, \tag{B3}$$

where $K$ has been defined in Sect. 2.2. Specifically, following to the results of Hillert (1965) and Ng (2016), we let

$$\bar{v} = \frac{K}{c_1 d_g} \,, \tag{B4}$$

where $c_1 \approx 2$ to 3. The assumption of normal grain growth gives a low-end estimate for $\bar{v}$, because strain-induced dynamic recrystallisation can accelerate grain boundary migration (Duval and Castelnau, 1995). On the other hand, impurities and

bubbles may reduce grain-boundary mobility (Alley, 1986a; 1986b). In the present formulation, we exclude these complications, as well as anisotropic vein motion resulting from recrystallisation processes. We ignore any influence on $K$ by the solute concentration $c$, because $c$ refers to impurities dissolved in vein water, rather than impurities at grain boundaries.

Next we calculate the transport fluxes. Imagine a region of uniform porosity and uniform mean grain size where the vein network continually evolves. The incremental flux of porosity (vein space) crossing an area $dA$ in the direction perpendicular to the plane is

$$dJ = \phi \, v\cos\theta' \, F(v)v^2 \sin\theta' \, d\theta' d\phi' dv dA .$$ (B5)

Integrating this over $v \in [0, \infty]$, $\phi' \in [0, 2\pi]$ and $\theta' \in [0, \pi/2]$ (for unidirectional flux; Fig. B1) yields the porosity flux density

$$j = \frac{dJ}{dA} = \phi\pi \int_0^\infty F(v)v^3 \, dv = \frac{1}{4}\overline{v}\phi .$$ (B6)

The same method applied to the bulk solute content $c_B = \phi c$ gives its flux density as

$$j_c = \frac{1}{4}\overline{v}c_B .$$ (B7)

Given how these macroscopic fluxes originate from microscopic interactions, they are valid on time scales longer than the time scale of vein-crossing events, $\sim d_g/\overline{v}$.

In a uniform region, the fluxes through $dA$ in opposite directions cancel. But if $\phi$ (or $c_B$) varies spatially, a differential 665 flux occurs; diffusion then arises from the vein motion. The net diffusion rate across a plane, say, at elevation $z = z_0$, is found by subtracting the opposite fluxes at a distance $dz$ on either side, $j_+$ at $z_0 - dz$ and $j_-$ at $z_0 - dz$, where $dz$ locates the planes for evaluating the fluxes from continuum properties. We determine $dz$ by using an argument similar to that in the kinetic theory. Moving vein segments merge and reconfigure on distances on the order of the mean grain size, so vein segments arriving at the plane typically come from a distance $\sim d_g$ since their last "collision", which caused them to switch direction and gain or 670 lose solute; $d_g$ is akin to the particle mean free path in the kinetic theory. Accordingly, we choose $dz$ to be the flux-averaged value of the perpendicular distance $d_g \cos\theta'$ :

$$dz = \frac{\int_0^\infty \int_0^{2\pi} \int_0^{\pi/2} d_g \cos\theta' F(v)v^3 \cos\theta' \sin\theta' \, d\theta' d\phi' dv}{\int_0^\infty \int_0^{2\pi} \int_0^{\pi/2} F(v)v^3 \cos\theta' \sin\theta' \, d\theta' d\phi' dv} = d_g \frac{\int_0^{\pi/2} \cos^2\theta' \sin\theta' \, d\theta'}{\int_0^{\pi/2} \cos\theta' \sin\theta' \, d\theta'} = \frac{2d_g}{3} .$$ (B8)

It follows that the net diffusive transport across the plane is

$$j_{net} = j_+ - j_- \;=\; \frac{1}{4}\,\overline{v}\,\left.\phi\right|_{z_0 - 2D/3} - \left.\phi\right|_{z_0 + 2D/3} \;\approx\; \frac{1}{4}\,\overline{v}\left(-\frac{4d_g}{3}\left.\frac{\partial\phi}{\partial z}\right|_{z_0}\right)$$

$$= -\frac{\overline{v}d_g}{3}\left.\frac{\partial\phi}{\partial z}\right|_{z_0}. \tag{B9}$$

Combining this result with (B4) gives the diffusivity $\kappa = \overline{v}d_g/3 = K(T)/3c_1$, as given in (9) in the text. Figure B2 plots $\kappa$ against temperature. Interestingly, $\kappa$ is independent of $d_g$ because smaller grains lead to faster grain boundary migration, but proportionally shorter mean free path for the vein motion.

### Code and data availability

Our MATLAB code and the simulated data of our control runs are archived at doi:10.15131/shef.data.12735191.
Please use https://figshare.com/s/8607e837455c5188c207 during the review stage.

### Supplement link

Movies S1–S8 and Figs. S1–S3 can be accessed via doi:10.15131/shef.data.12739169.
Please use https://figshare.com/s/aa059ab52b73f472f3fd during the review stage.

### Author contribution

F. S. L. Ng designed and performed the study and wrote the paper.

### Competing interests

The author declares that he has no conflict of interest.

### Acknowledgements

I thank R. Traversi for providing the data in Fig. 1a and 1b; A. J. Sole and A. J. Hepburn for comments on the pre-submission manuscript; and an anonymous reviewer, A. Rempel and E. Wolff for valuable suggestions.

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

**Table 1:** Constants used in this study.

| Symbol | Value | Parameter |
| --- | --- | --- |
| $\alpha$ | 0.0725 | Geometric constant |
| $c_1$ | 2.5 | Geometric constant |
| $D$ | *$5 \times 10^{-10}$ m$^2$ s$^{-1}$ | Solute diffusivity in vein water |
| $\gamma$ | 0.034 J m$^{-2}$ | Interfacial energy (Gibbs–Thomson coefficient) |
| $\Gamma$ | 4.53 K M$^{-1}$ | Slope of water–SO$_4^{2-}$ liquidus curve, $\approx$ 0 °C (Dani et al., 2012) |
| $g$ | 9.81 m s$^{-2}$ | Gravitational acceleration |
| $K_0$ | **$1.68 \times 10^7$ mm$^2$ yr$^{-1}$ | Grain growth rate constant |
| $L$ | $333.5 \times 10^3$ J kg$^{-1}$ | Latent heat of melting |
| $\rho_i$ | 917 kg m$^{-3}$ | Ice density |
| $\rho_w$ | 1000 kg m$^{-3}$ | Water density |
| $Q$ | 42.4 kJ mol$^{-1}$ | Grain growth activation energy |
| $R$ | 8.314 J K$^{-1}$ mol$^{-1}$ | Gas constant |
| $T_0$ | 273.15 K (0 °C) | Reference temperature (melting point at 1 bar) |

* One-third of the molecular diffusivity in water; see Rempel et al. (2001).

** Value derived from Table 3.1 of Cuffey and Paterson (2010), after multiplying by $6/\pi$ to correct for sectioning and stereological effects

(Ng and Jacka, 2014)



**Table 2:** Variables in our mathematical model and their units.

| Symbol | Physical meaning, unit |
|---|---|
| $c$ | Impurity concentration of the vein water, M |
| $c_B$ | Bulk impurity concentration (vein-water component only), M |
| $d_g$ | Mean grain size (diameter), m |
| $g(t)$ | Depth-age scale (i.e., depth of ice having the age $t$), m |
| $H$ | Ice thickness, m |
| $K$ | Grain growth rate, $m^2$ $yr^{-1}$ |
| $l$ | Vein length density, $m^{-2}$ |
| $m$ | Melt rate (mass rate per unit volume of ice), kg $m^{-3}$ $yr^{-1}$ |
| $\mathbf{q}$ | Vectorial percolative water flux in ice (scalar, $q$), m $yr^{-1}$ |
| $r_v$ | Radius of curvature of vein faces, m |
| $t$ | Time, yr (when referring to age in an ice core, a is used) |
| $T$ | Temperature, K or °C |
| $\mathbf{u}$ | Ice velocity vector, m $yr^{-1}$ |
| $\mathbf{u}_c$ | Anomalous velocity vector (scalar, $w_c$), m $yr^{-1}$ |
| $\tilde{w}$ | Ice velocity in a reference frame following ice material, m $yr^{-1}$ |
| $\mathbf{x}$ ($= (x, y, z)$) | Cartesian coordinate in 3D, m (see next item for orientation) |
| $z$ | Depth below ice-sheet surface, m |
| $z'$ | Vertical displacement in a reference frame following ice material, m |
| $\Delta$ | Width parameter of Gaussian function, m |
| $\theta_c$ ($/\theta_p$) | Melting-point depression due to impurity (/pressure), K |
| $\kappa$ | Residual diffusivity due to random vein motion, $m^2$ $yr^{-1}$ |
| $\phi$ | Porosity, dimensionless |

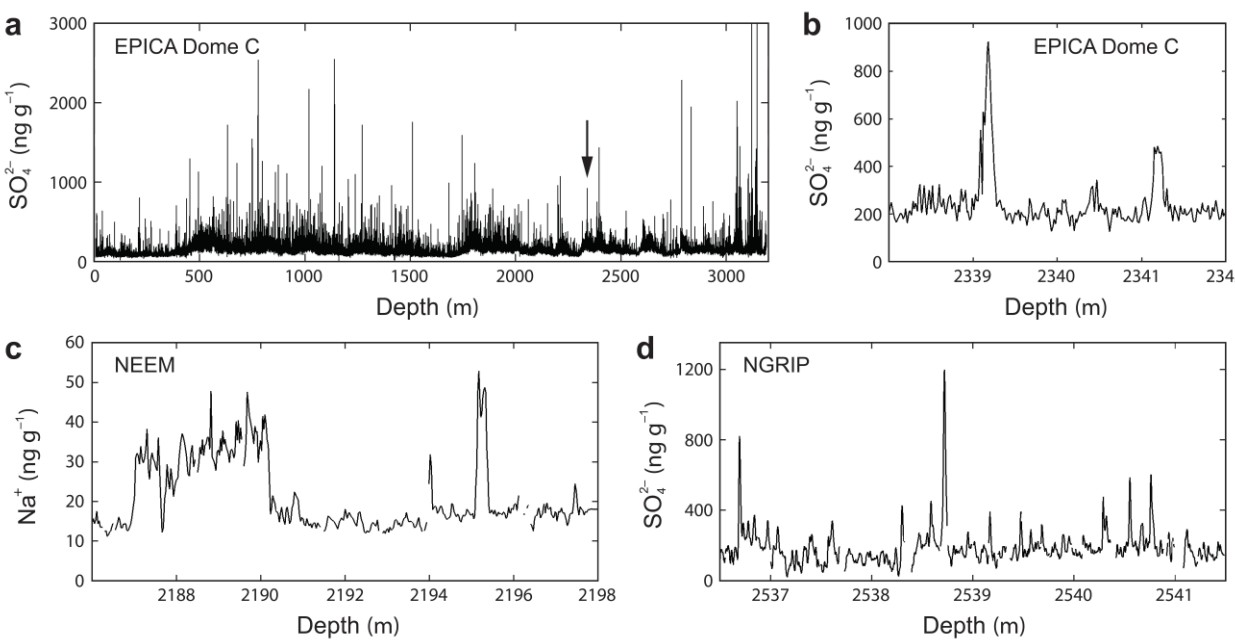

**Figure 1**: High-resolution records of dissolved ionic impurities in ice cores from Antarctica and Greenland, illustrating the occurrence and expressions of high peaks and signals at decimetre or shorter scale in deep ice. Vertical axes plot the total impurity concentration in bulk ice (this differs from our model variable $c_B$, which refers only to impurities in the water-vein network); 1 ng g$^{-1}$ is equivalent to 1 ppb or 1 $\mu$g L$^{-1}$. (a) The sulphate record from the EPICA Dome C core, Antarctica, measured by fast ion chromatography (Traversi et al., 2009); the arrow locates panel b. (b) Zoomed portion of (a) showing two sulphate peaks. (c) A piece of the sodium record from the NEEM ice core, Greenland (Schüpbach et al., 2018) from continuous flow analysis (CFA) measurements, showing large fluctuations on a stretch with high base level in 2187–2190 m as well as isolated peaks. (d) A piece of the sulphate record from the NGRIP ice core, Greenland (Svensson et al., 2013) measured by CFA, showing successive sub-decimetre-scale spikes. Gaps in c and d reflect missing data.





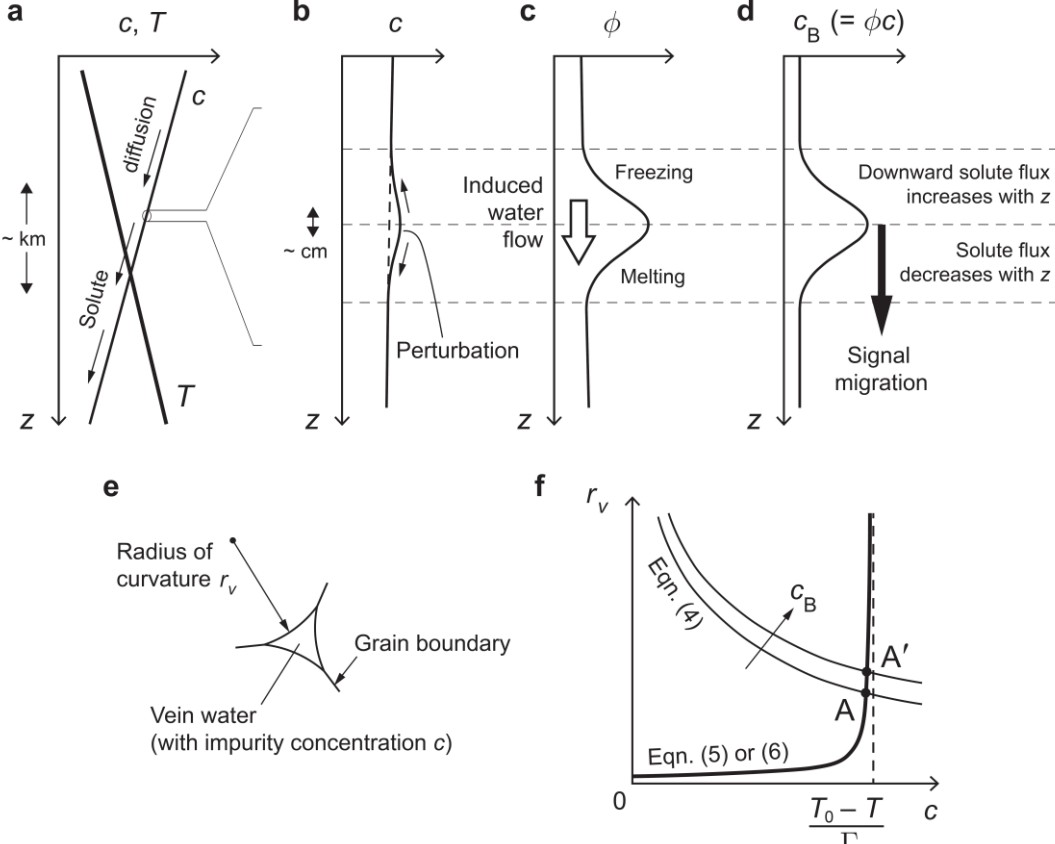

**Figure 2:** Interactions that cause the signal migration mechanism of Rempel et al. (2001) and our signal diffusion mechanism, and variables studied in this paper. (a) Conceptualisation of the vertical profiles of vein impurity concentration $c$ and ice temperature $T$ in an ice core; $z$ is depth below the surface. The gradients shown here typically occur over the kilometre length scale and are most pronounced towards the base of an ice sheet; see Fig. 4 for real examples of $T(z)$. Panels b, c and d expand on the variations around a $c_B$ signal, which occur at a much shorter scale of decimetres or centimetres. (b) Vein impurity concentration $c$. (c) Porosity $\phi$. (d) Bulk impurity concentration $c_B$. (e) Vein cross-sectional geometry, showing the definition of the radius of curvature $r_v$. (f) Equilibrium vein conditions as the solution of model equations. An increase in $c_B$ lifts the solution A to A', causing a perturbation on $c$ (see panel b) that drives $c_B$ diffusion. Neglect of the Gibbs–Thomson effect in the theory of Rempel et al. (2001) causes the bold curve in (f) to collapse onto the vertical dash line; then no perturbation arises to diffuse the signal.

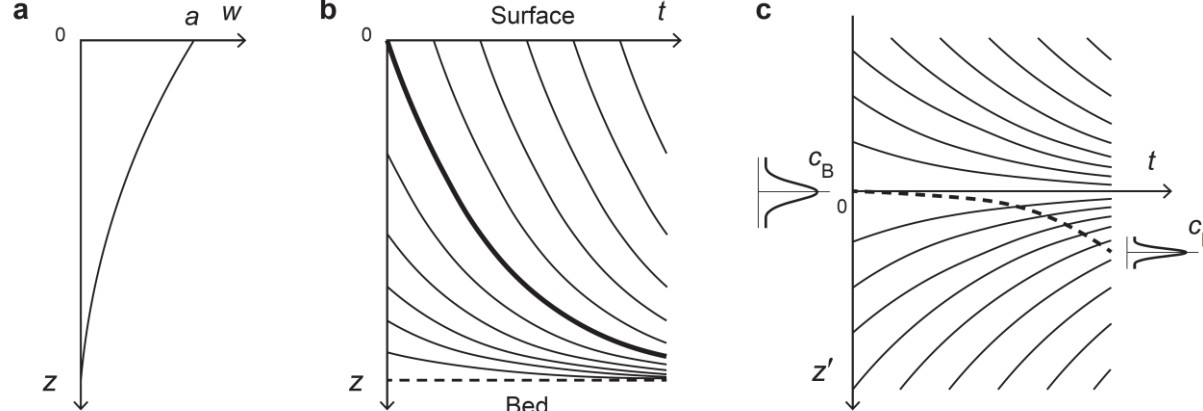

**Figure 3:** (a) Submergence velocity $w(z)$ in an ice core and the corresponding (b, c) $z$–$t$ and $z'$–$t$ plots of material trajectories. The bold line in (b) is the depth-age relation. In (c), $z'$ is a depth coordinate that measures distance from ice located on the depth-age relation in (b). The $z'$-reference frame thus moves with that ice as it descends towards the bed, and material trajectories that do not start at the surface at $t = 0$ converge towards $z' = 0$ due to vertical compression. An impurity signal centred initially at $z' = 0$ can migrate by anomalous diffusion into $z' > 0$ (dashed curve) and diffuse, while it experiences vertical compression. The $z'$ coordinate system is used in the signal evolution experiments reported in Figs. 5 to 12.





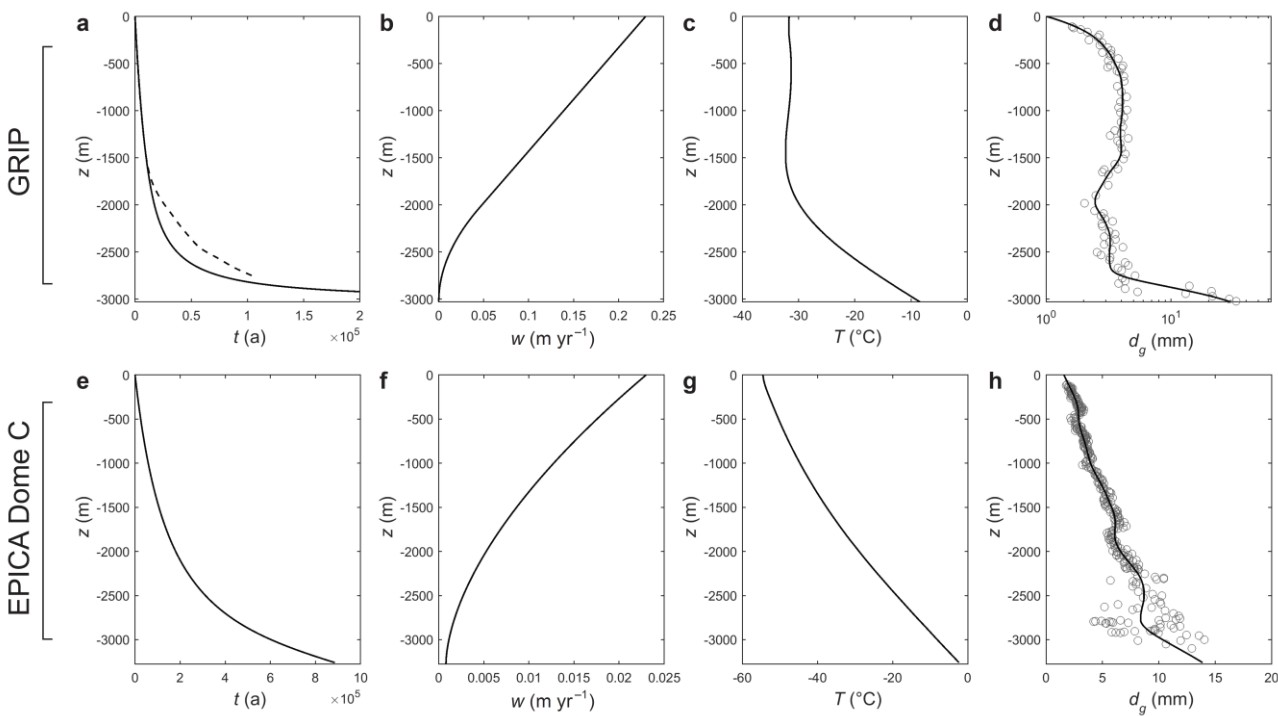

**Figure 4:** Ice-core background fields used in our signal evolution experiments. GRIP ice core: a to d. (a) Age-depth scale and (b) ice velocity from a Dansgaard-Johnsen model. Dashed line in (a) shows the GICC05 modelext time scale (Seierstad et al., 2014; Rasmussen et al., 2014). (c) Ice temperature from Johnsen et al. (1995). (d) Grain-size data from Thorsteinsson et al. (1997) and spline fit used in our modelling. EPICA Dome C core: e to h. (e) Age-depth scale and (f) ice velocity from the model described in Sect. 3.1. (g) Borehole temperature from Pol et al. (2010). (h) Grain-size data from Durand et al. (2004) and spline fit used in our modelling.




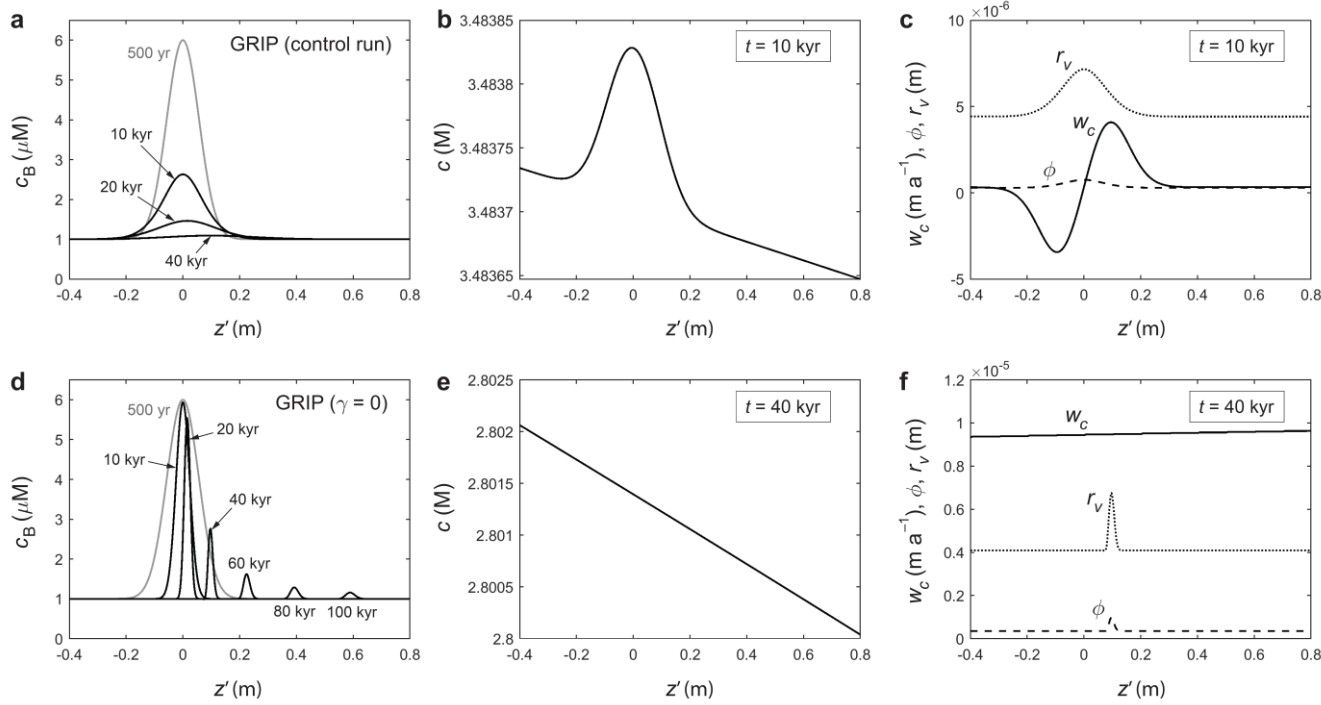


**Figure 5:** Modelled evolution of a signal peak in (a–c) the GRIP control run and (d–f) an otherwise identical run where the Gibbs–Thomson effect is turned off ($\gamma = 0$). Snapshots are shown in the material reference frame, with displacement $z'$ measuring how far the signal has moved from ice of the same age (which lies at $z' = 0$). (a,d) bulk solute concentration $c_B$; (b,e) vein solute concentration $c$ at one time; (c,f) anomalous velocity $w_c$, porosity $\phi$, vein curvature $r_v$ at one time. Grey curves in (a) and (d) indicate the initial doped peak. Panel b illustrates

the Gibbs–Thomson perturbation. See Movie S1 for the full simulations.


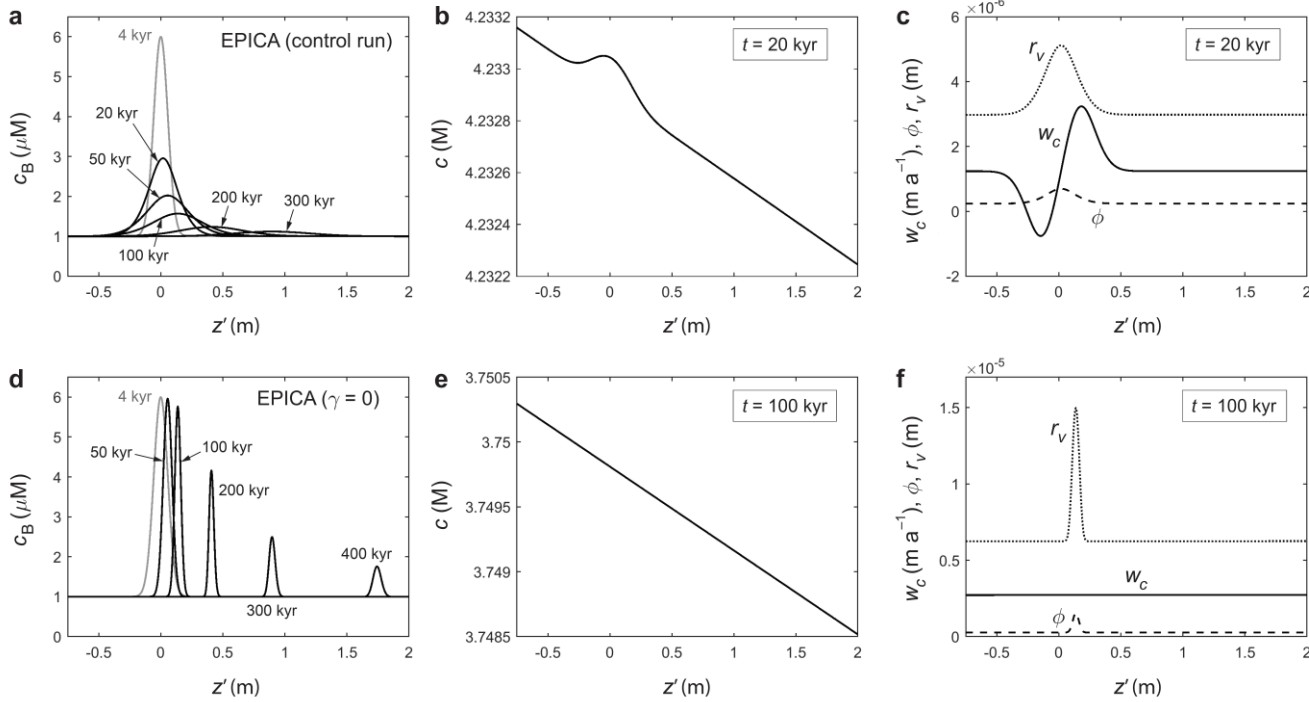

**Figure 6:** Modelled evolution of a signal peak in (a–c) the EPICA control run and (d–f) an otherwise identical run where the Gibbs–Thomson effect is turned off ($\gamma = 0$). (a,d) bulk solute concentration $c_B$; (b,e) vein solute concentration $c$ at one time; (c,f) anomalous velocity $w_c$, porosity $\phi$, vein curvature $r_v$ at one time. Grey curves in (a) and (d) indicate the initial doped peak. Panel b illustrates the Gibbs–Thomson perturbation. See Movie S2 for the full simulations.


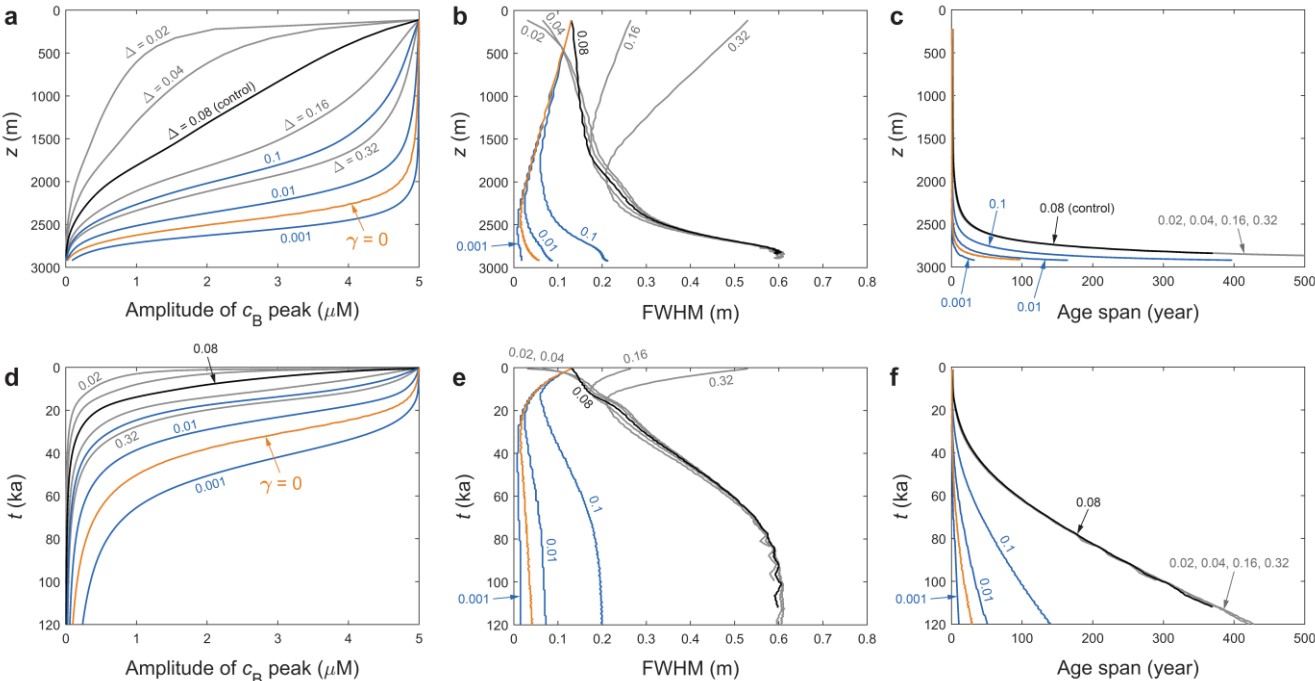

**Figure 7:** Changing morphometry of the signal peak – its amplitude, full width at half maximum (FWHM), age span – in the GRIP ice core for different model parameters, plotted against depth (a–c) and age of the ice (d–f). Black curves plot the control run of Fig. 5a, and orange

curves the $\gamma = 0$ run of Fig. 5d. Grey curves plot the results of altering the width parameter $\Delta$ of the doped peak from 0.08 (control) to four other values. Blue curves plot the outcomes of suppressing molecular diffusivity $D$ in the control run by the multiplicative factors 0.1, 0.01 and 0.001, to simulate vein blockage. Parameter labels use the same colours as the curves. Peak width becomes difficult to measure as amplitude diminishes, explaining the jittery appearance of some curves at depth.



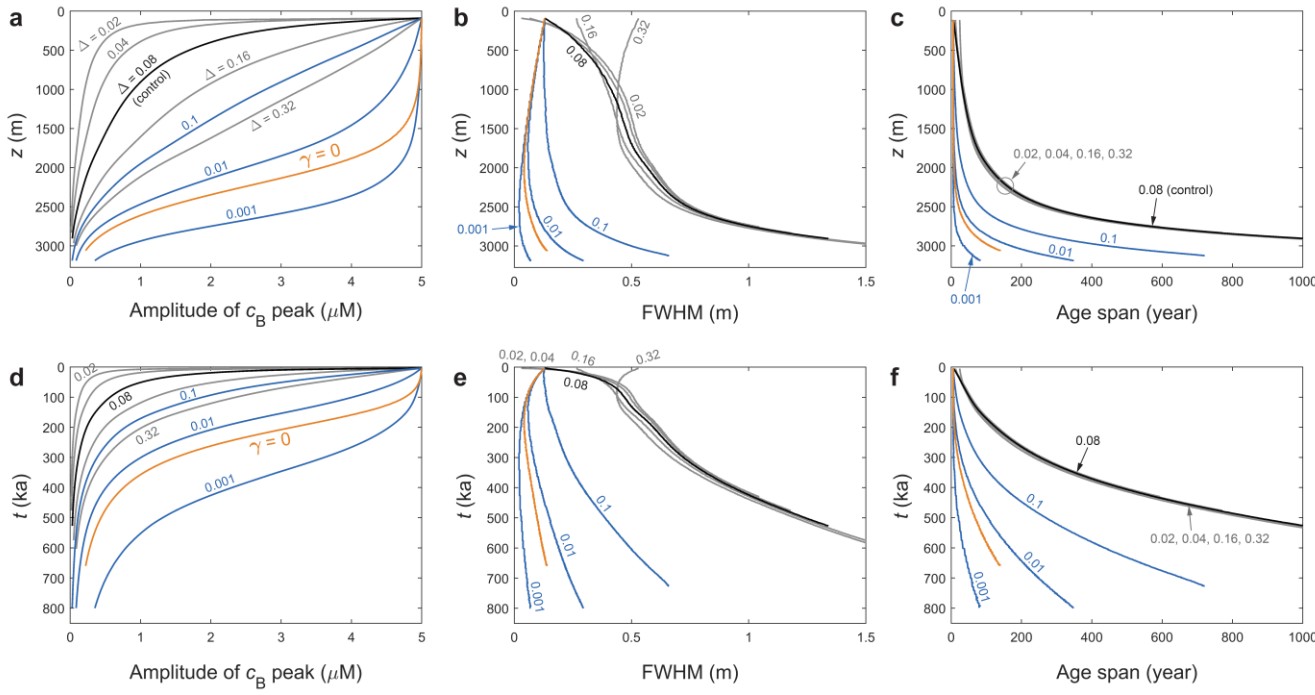


**Figure 8:** Changing morphometry of the signal peak – its amplitude, full width at half maximum (FWHM), age span – in the EPICA ice core for different model parameters, plotted against depth (a–c) and age of the ice (d–f). Black curves plot the control run of Fig. 6a, and orange curves the $\gamma = 0$ run of Fig. 6d. Grey and blue curves document the same sensitivity tests as conducted for the GRIP core (see Fig. 7 caption for details). The FWHM and age-span axes are scaled to focus more on the blue and orange curves, rather than the deep ends of the grey/black curves, as the corresponding signal amplitudes decay to near zero.



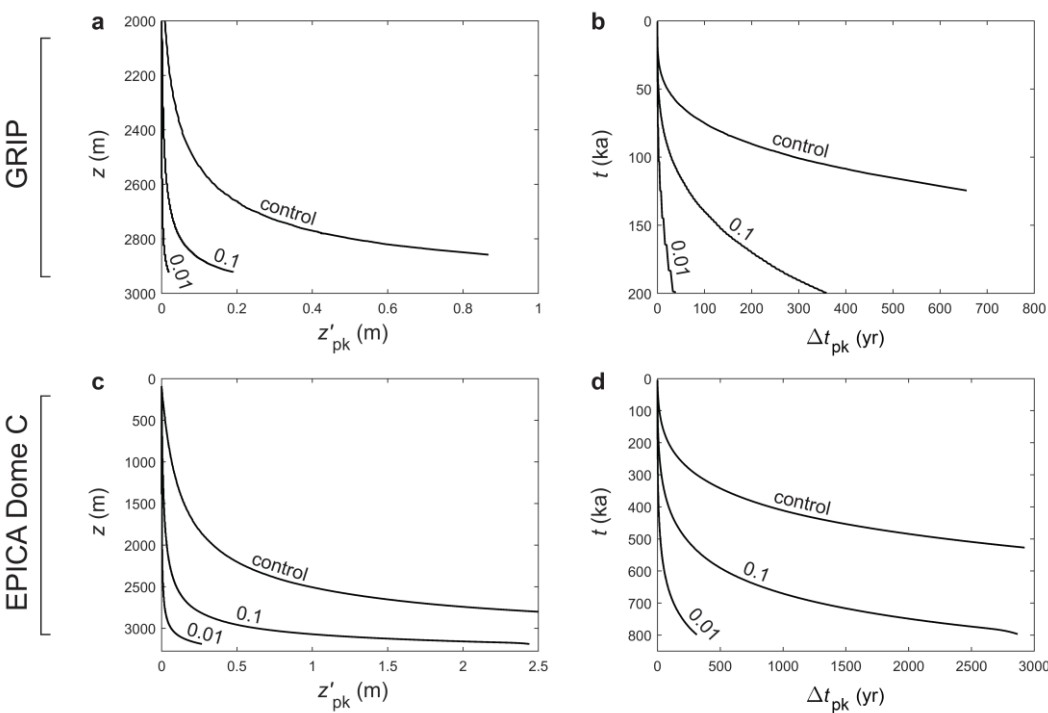

**Figure 9:** Modelled displacement $z'_{pk}$ and age offset $\Delta t_{pk}$ of signal peaks (from ice of the same age) at the (a,b) GRIP and (c,d) EPICA core sites for different parameters, plotted against depth and age of the ice. "Control" labels the control runs in Figs. 5a and 6a; 0.1 and 0.01 label those runs in Figs. 7 and 8 where the molecular diffusivity $D$ is suppressed by these factors to simulate vein blockage. The control curve in panel a is equivalent to the curve in Fig. 4 of Rempel et al. (2001), except these authors assumed a different age-depth scale from ours.




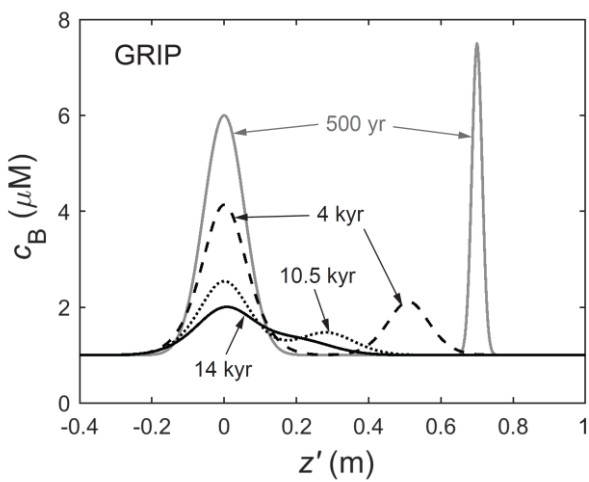


**Figure 10:** Snapshots (at four times) of the evolution of two neighbouring peaks in a GRIP run that uses the control parameters of the run in Fig. 5a. Diffusional spreading causes the peaks to merge as they approach each other under vertical compression. See Movie S3 for the full simulation.





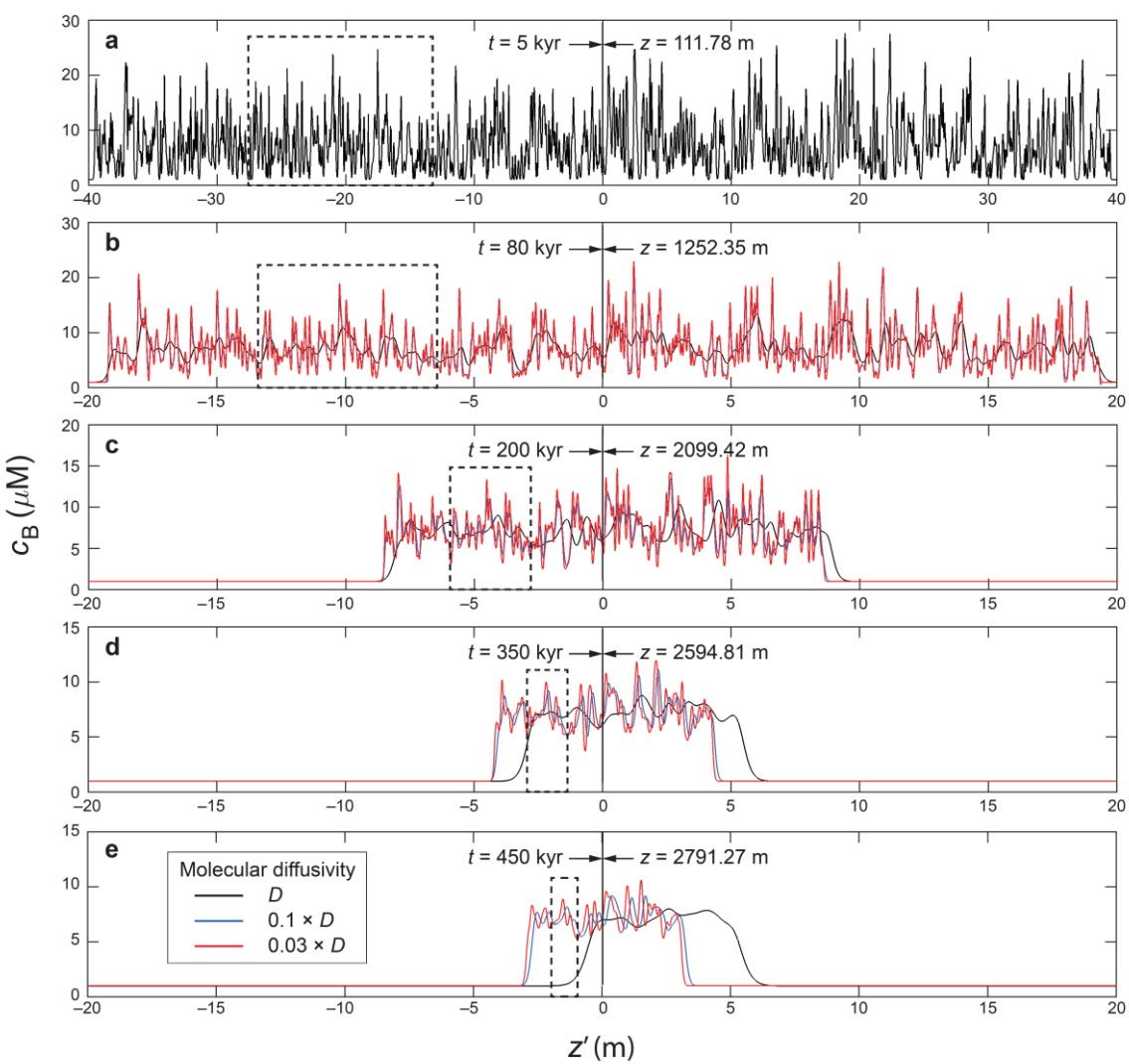

**Figure 11:** Simulated evolution of a long sequence of doped signals at the EPICA site, in three experiments assuming the molecular diffusivities $D$ (Table 1), $0.1D$ and $0.03D$. (a) Initial $c_B$ sequence. Used in all three runs and made from the superposition of 1200 decimetre-scale Gaussian peaks, it is not meant to recreate the actual signals at EPICA. (b)–(e) Snapshots of $c_B$ at later times. Labels near the vertical line indicate the depth and age of the ice at $z' = 0$. The dashed boxes trace a group of signals as they evolve into new signals through compression-diffusion merging (Sect. 3.3). As $D$ is reduced, signal persistence into deep ice improves, and signal displacement decreases. The duration of the signal sequence is $\approx 3,600$ yr in all panels. See Movie S5 for the full simulations.


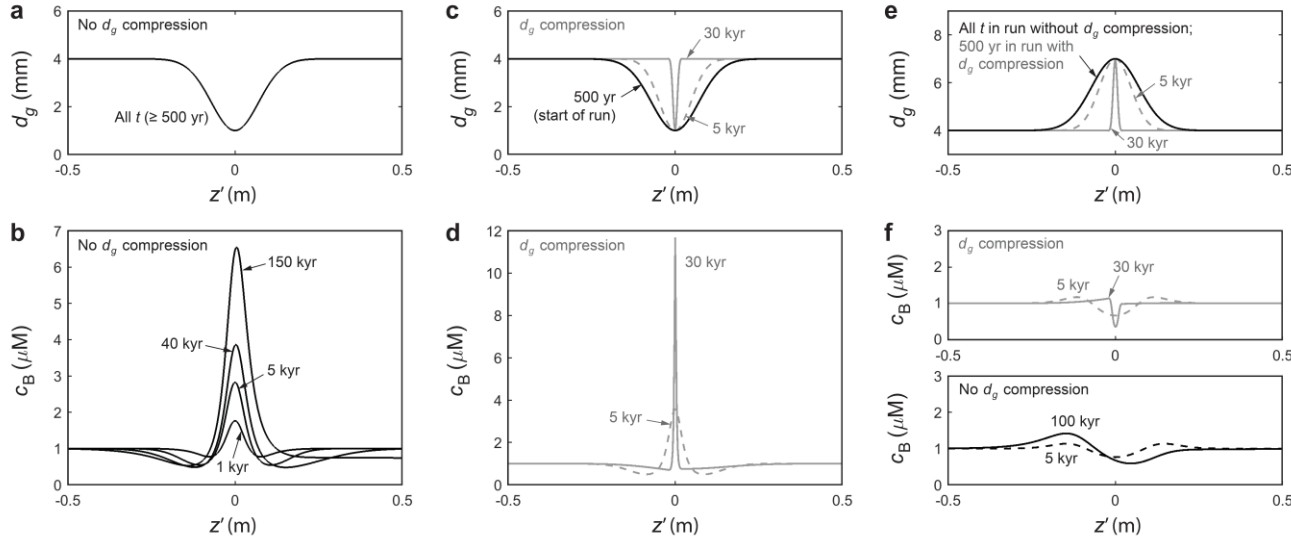

**Figure 12:** Four modified GRIP control runs, demonstrating how fluctuations in the mean grain size $d_g$ (a, c, e) at decimetre scale cause signals to form on the bulk concentration $c_B$ (b, d, f). All runs begin at $t = 500$ yr with $c_B \equiv 1$ $\mu$M without an initial impurity signal. Evolution

snapshots are shown in the material reference frame. (a, b) Experiment imposing a negative $d_g$ fluctuation of a fixed width. (c, d) Experiment imposing a negative $d_g$ fluctuation that has the same initial form as in (a), but which narrows due to vertical compression of the ice. (e, f) Two experiments with a positive $d_g$ fluctuation, set up as in the last two experiments. One run assumes that the fluctuation does not experience compression (black); the other run assumes that it does (grey). In those runs where the $d_g$ fluctuation is compressed, only results up to 30 kyr are shown, because soon afterwards its width becomes too narrow to be resolved by the numerical grid spacing (0.0025 m). In all runs,

the new signal in $c_B$ is localised by the grain-size fluctuation and does not displace into $z' > 0$ by anomalous diffusion. See Movie S6 for the full simulations. Fig. S3 and Movie S7 present the equivalent experiments for the EPICA core site.


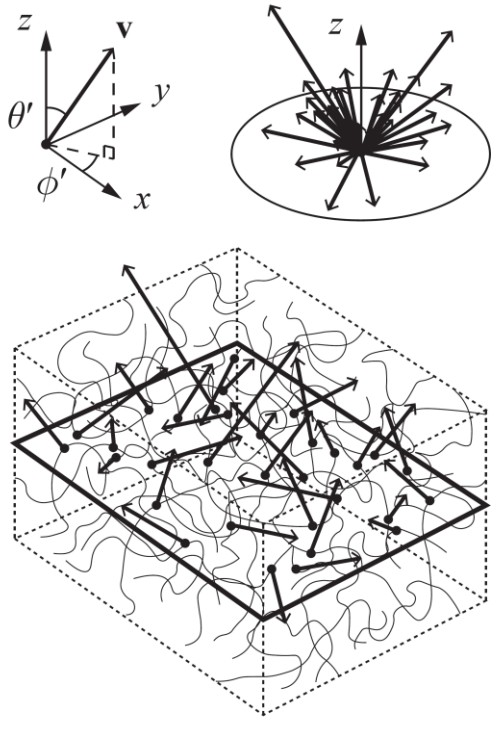

**Figure B1:** Velocities **v** of different vein segments crossing a plane (outlined by bold rectangle), and their distribution in spherical coordinates (upper right). Only crossings in one direction are shown. In Appendix B, a statistical theory is used to calculate the net transport of porosity and vein impurity resulting from this motion.

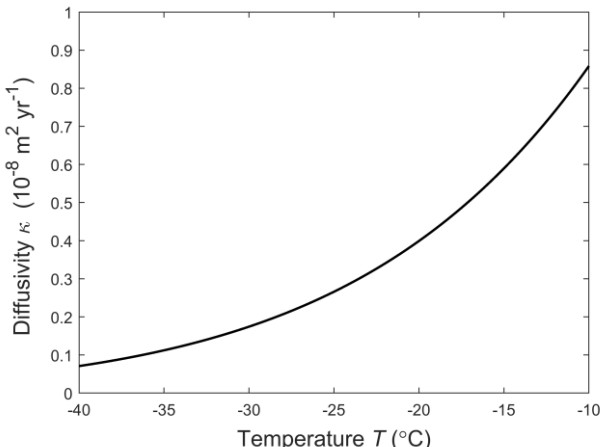

**Figure B2:** Diffusivity $\kappa$ at different temperatures, calculated with (9) for $c_1 = 2.5$.

1120