# Peer review of "Pervasive diffusion of climate signals recorded in ice-vein ionic impurities"

_The Cryosphere, 2020_

## Referee Comment (RC1) · Anonymous Referee #1 · 2 Oct 2020

Overall assessment

The paper by Ng tackles an issue from a theoretical point of view that bothered ice core science for nearly two decades, i.e., the potential signal migration of bulk chemical signals through the vein network in the presence of an in situ temperature gradient. This hypothesis was put forward by A. Rempel in 2001, but while no unambiguous ice core evidence for this hypothesis was provided in the meantime it could also not be refuted as the theory of migration through the vein network is physically not disputed. However it is still unknown how much of the impurities are located in veins and, thus, really subject to this transport.

The results by Ng provide a convincing argument why no evidence for a migrated signal CAN be found, i.e., because any migration signal would also be subject to intense

diffusional smoothing. Ng expands on the theory by Rempel by taking into account the curvature effect on vein volume that had been intentionally neglected by Rempel. Ng is able to convincingly show that taking this additional term into account, does not stop any migration but leads to strong diffusion of such a migrating signal and, thus, to the disappearance of the displaced signal in the ice. This has two very important implications for ice core science: Firstly, any short-lived peaks in the ice core record that are imprinted in the veins cannot survive the migration process. Vice versa, if there is a peak found in the ice core record it cannot be shifted relative to its initial bulk position. Secondly, the presence of distinct peaks in many deep ice core records suggests that vein migration and diffusion cannot be a dominant process as otherwise any peaks would have disappeared. This implies, as explicitly outlined in the paper by Ng, that either the part of impurities located in veins is only small or that vein transport is increasingly suppressed by a fragmenting vein network as already suggested by other authors.

The paper is well written and excels by its stringent, mathematical approach, while at the same time performing model experiments that are very instructive and, therefore, of great value for the ice core practitioners. It is therefore of high relevance and well suited for publication in The Cryosphere. The mathematical approach (although well laid out) and some for the outsider unintuitive formulations make the access to the paper a little more difficult than necessary. Below I make some suggestions for minor revisions that may help to remedy that. In summary, I highly recommend the paper for publication in The Cryosphere after taking care of these minor revisions.

General comments

The paper is a little "dry" as there are no ice core examples provided. Thus, for those readers who are not specialists in ice core chemistry, it is hard to imagine how those signal that Ng talks about, look like. Here, it would be helpful to see (in the Introduction or the Discussion) a figure that shows examples for high peaks in deep ice on the one hand (for example for in the $Ca^{2+}$ or $SO_4^{2-}$ record) and examples of a general

reduction of variability as seen on others (for example in the Na+ and NO3- record). Examples can be found in the papers by (Traversi et al., ES&T 2009, Röthlisberger et al., CP 2008, Schüpbach et al., Nature Communications 2018, Barnes et al., JGR 2003, and others).

In addition to Table 1, which provides the values of the constants used, I would suggest to provide a table 2 where all variables used are listed with a short explanation and their units.

The information on the age span is provided in figures 6 and 7, however the discussion of this parameter, despite its great importance for ice core science in particular for very old ice, is rather limited. Looking at the EPICA Dome C results it appears that the age span in Fig. 7 approaches several thousand years or even the precession age scale when reaching an age of 700-800 kyr (unfortunately the current scale of the x-axis does not allow to quantify this for the control run). On the other hand, the EPICA Dome C ice core record provides some information about the time scale of variability that can still be resolved, thus, would constrain the degree of vein transport smoothing empirically. I would recommend to extend the discussion on this point, as it is of great relevance for ice core sciences.

Specific comments:

Introduction 1st paragraph. You could also mention here already the loss of species such as NO3- or Cl- at the surface (Röthlisberger et al., Ann Glac 2002, Weller et al., JGR 2004) migration processes (for example for methanesulphonate) that occur already at the surface (Osman et al., 2017) or the aggregation of dust particles in the deepest ice (Tison et al., 2015)

p2 l33: "...migrate relative to the ice..."

p2 l35: "... could decouple..."

p2 l41: "... signal migration in deep ice may..."

p2 l51: this is one of the examples where the author refers to the ice core evidence without showing it

p2 l62: One intrinsic assumption made in the Rempel theory and also in the work by Ng is that $c\_B=c*Phi$. However, if impurities are located in microinclusions or grain boundaries, hence not in contact with the veins, then the vein transport is not representative of changes in bulk concentrations. In fact, this is discussed later, but I would recommend that this assumption is explicitly mentioned when $c\_B=c*Phi$ is introduced.

p3 l63: when I first read the word "ice porosity" I got confused. I would suggest to write: " is mirrored by variations in the liquid filled vein volume relative to the total ice volume, in the following called ice porosity Phi"

p3 line 68: please change the unclear wording "Thus the $c\_B$ peak translates"

p4 l96: replace "we" by "I" throughout the manuscript

p4 l111: "The three terms on its right-hand side describe the temperature depressions due to (i) solute, (ii) interfacial curvature (the Gibbs-Thomson effect) and (iii) pressure, respectively;"

p4 l 113: what does the constant gamma represent?

p6 l147: "... where the melt rate m (in units?) at the interfacial boundaries of the vein accounts..."

p6 line 155: why not explain the last term here?

p7 equations 17 and 18: You neglect dw/dz. Say explicitly why.

p11 l256: "while anomalous (Rempel) diffusion"

p11 l273-274: use a different variable for the basal melt rate (for example m_bas) to distinguish it from the interfacial melt rate m used above

p12 l298: Likely a topic to be looked at in a separate paper, but the theory by Ng

should be also able to predict how much vein diffusional smoothing occurs for the water isotopes.

p13 l318: " to become so large to be ..."

p13 l321. The data provided in the Mayewski papers are not of really high resolution. You may also want to cite Schüpbach et al, Nature Communications 2018, Bigler et al., Quaternary Science Reviews 2010, Röthlisberger et al, CP 2008)

p13 l332: "we observe an interesting"

p13 l335-337: expand the discussion of the age span and the potential resolution loss

p15 3rd paragraph: The movies provided are very helpful! In Movie S3 and Figure 9 one of the two peaks in the GRIP ice core moves relatively upward. This deserves some discussion in the main text. Also the y-axis for the temperature profile in the movies could be scaled equally for all time steps.

p16 l403-404. I am not sure I understand this correctly, please clarify. Again, a discussion of the time-scale that could be resolved and those found in ice core records would be helpful.

p17 scenario 2: here you mention the issue of micro-inclusions which would contradict the assumption that $c_B = c*Phi$. As mentioned before, this assumption should be qualified as such earlier in the manuscript.

p17 l455: "vein $c_B$" is the wording correct???

p17 l458-459. The sentence "In any case... from $SO_4^{2-}$" is way too general and should be specified. Fr example to which depth could we see volcanic peaks at GRIP and EPICA Dome C. Which longer-term variations could be still resolved (discussion of age span)

p18 l2: "of any vein impurities"

p18 line 476-478. expand the discussion on longer variations

p19 l498-500: This comes unexpected and justifies a little bit more discussion.

Figure 1: I know these are only illustrative figures, but the scale of the anomaly (centimeters) and the approximate location within the ice sheet of the sketch (lower third, where temperature gradients exist both in Greenland and Antarctica) should be indicated either in the figure itself or at least in the caption.

Figure 2: clarify what you mean by "that measures distance from ice at age t"

Figure 7: rescale panel f to show the age span for ice with an age of 800 kyr.

---

## Referee Comment (RC2) · Alan Rempel (Referee) · 9 Oct 2020

The manuscript by Felix Ng revisits the question of what happens to soluble impurities in glacial ice over the long time periods represented by ice-core climate records. Assuming that these impurities are primarily contained within a connected liquid-filled vein-node network along three- and four-grain contacts (e.g. Nye and Frank, 1973), an earlier model predicted that the control of impurity loading on liquid content should produce spatial variations in the effective compositional diffusivity that would help to explain the long-term preservation of short wavelength (e.g. representing seasonal to decadal time periods) bulk compositional signals, while also leading to their translation relative to the surrounding ice (Rempel et al., 2001). This so-called "anomalous diffusion" phenomenon has proved very difficult to test, since the predicted rate of signal

migration is very low (largely controlled by the shallow temperature gradients in polar ice) and firm constraints are rarely, if ever, available on the relative timing of compositional and ice-borne (e.g. oxygen isotopes) proxy deposition in the distant past. If the compositional signals themselves aren't altered in form, but only displaced a small distance relative to the ice with which they were originally deposited, what evidence is there to unequivocally demonstrate such subtle effects?

Ng's analysis predicts qualitatively different behavior in a slightly modified system, by focussing on the role of vein surface energy embodied in the Gibbs-Thomson effect, while assuming that the ice is characterized by locally uniform (but slowly growing) grain sizes. The models of anomalous diffusion that had been presented earlier had differed by employing a different assumption concerning the relationship between ice grain size and impurity loading. Measurements show that average grain size and impurity content are negatively correlated in ice cores (e.g., Gow and Williamson, 1976; Gow et al., 1997; Lipenkov et al., 1989; Azuma et al., 1999, 2000; Thorsteinsson et al., 1995; Durand et al., 2009). Rempel et al. (2001) highlighted this anticorrelation as consistent with the assumption that "the surface energy of curved interfaces acts to make vein radii uniform (Nye, 1989; Mader, 1992)", reasoning that "variations in [bulk impurity loading] must correlate with changes in the total length of veins per unit sample volume". Effectively, the idea was that by responding to impurity loading, grain sizes could adjust and prevent differences in vein radii from occuring. By contrast, Ng treats the grain size and impurity loading as unrelated, implying that vein radii are initially in disequilibium when changes in deposition produce changes in impurity loading, so that the vein radii subsequently evolve towards uniformity. This equilibration of vein radii drives additional computational transport that adjusts the impurity loading, thereby modifying the form of the compositional signals and preventing their displacement relative to the ice.

The analysis by Ng is elegant and clearly presented, and represents a welcome addition to the literature that describes post depositional processes with potential to be

relevant to ice core records. To better convey the departure from the earlier efforts to address this particular problem, it would be helpful to clarify the discussion surrounding the different assumptions regarding ice grain size. At present, the reason given by Ng for neglect of the Gibbs-Thomson effect in the Rempel et al. (2001) model is the scaling in the liquidus relation (line 122). However, the original reasoning provided by Rempel et al. (2001, p. 370) emerges from the assumption that an anti-correlation between grain size and impurity loading makes vein radii uniform, as predicted if grain size scales inversely with the square of bulk impurity loading in equation (7) of the current work, thereby rendering the second term in equation (8) spatially uniform as well. Unfortunately, as noted by Ng around line 239, "reliable grain-size modelling remains out of reach", though impressive improvements have been made in recent years (e.g. Faria et al., 2014; Ng and Jacka, 2014). Nevertheless, no convincing treatment has yet been provided that is quantitatively successful at representing the causal mechanisms that produce the observed anti-correlation between grain size and impurity loading. The current work highlights yet another reason, in addition to more commonly evoked considerations of the effects of grain growth on rheology and fabric development, for the importance of addressing this challenge.

Under the impurity-independent grain size assumption, Ng demonstrates convincingly that short-wavelength compositional variations (with impurities confined entirely to the vein-node network) should not persist over multi-millennial time scales under typical ice core conditions. Clearly, the apparent preservation of such signals deep in the ice requires further explanation, and Ng provides considerable insight into various potential mechanisms that might resolve this conundrum. Ultimately, two scenarios are championed, with the first involving some mechanism for blocking vein transport, for example with dust particles (e.g. Raymond and Harrison, 1975), and the second relying on impurities being largely confined to the ice matrix or two-grain boundaries (e.g. Eichler et al., 2017). The model of Rempel et al. (2001) shows that a strong coupling between bulk impurity content and grain growth could provide a third mechanism by reducing or eliminating spatial variations in vein radius and greatly damping the influence

of the Gibbs-Thomson effect on impurity transport. Likely some elements of all three scenarios contribute, and efforts to quantify post-depositional changes would benefit from further efforts aimed at unravelling their relative importance. Ng's manuscript represents a valuable, detailed exploration of the end member case in which grain size is independent of bulk impurity content. This clear and cogent analysis has important implications, and it is to be hoped that it will motivate efforts that provide even further constraints on how exactly post depositional compositional migration occurs in nature.

I note that Reviewer 1 has already provided detailed suggestions for minor wording changes and I have nothing significant to add.

References:

Azuma, N., Wang, Y., Mori, K., Narita, H., Hondoh, T., Shoji, H., and Watanabe, O.: Textures and fabrics in Dome F (Antarctica) ice core, Ann. Glaciol., 29, 163–168, doi:10.3189/172756499781821148, 1999.

Azuma, Nobuhiko, et al. "Crystallographic analysis of the Dome Fuji ice core." Physics of Ice Core Records. Hokkaido University Press, 2000.

Durand, G., Svensson, A., Kipfstuhl, S., Persson, A., Gagliardini, O., Gillet, F., Sjolte, J., Montagnat, M., and Dahl-Jensen, D.: Evolution of the texture along the EPICA dome C ice core, Insti- tute of Low Temperature Science, Hokkaido University, Sapporo Japan, Nepal, vol. 68, 91–106, 2009.

Eichler, Jan, et al. "Location and distribution of micro-inclusions in the EDML and NEEM ice cores using optical microscopy and in situ Raman spectroscopy." The Cryosphere 11.3 (2017): 1075-1090.

Faria, Sérgio H., Ilka Weikusat, and Nobuhiko Azuma. "The microstructure of polar ice. Part II: State of the art." Journal of structural geology 61 (2014): 21-49.

Gow, Anthony J., and Terrence Williamson. "Rheological implications of the internal structure and crystal fabrics of the West Antarctic ice sheet as revealed by deep core

drilling at Byrd Station." Geological Society of America Bulletin 87.12 (1976): 1665-1677.

Gow, A. J., et al. "Physical and structural properties of the Greenland Ice Sheet Project 2 ice core: A review." Journal of Geophysical Research: Oceans 102.C12 (1997): 26559-26575.

Lipenkov, V. Ya, et al. "Crystalline texture of the 2083 m ice core at Vostok Station, Antarctica." Journal of Glaciology 35.121 (1989): 392-398.

Mader, Heidy M. "Observations of the water-vein system in polycrystalline ice." Journal of Glaciology 38.130 (1992): 333-347.

Ng, Felix S.L., and T. H. Jacka. "A model of crystal-size evolution in polar ice masses." Journal of Glaciology 60.221 (2014): 463-477.

Nye, J. F. "The geometry of water veins and nodes in polycrystalline ice." Journal of Glaciology 35.119 (1989): 17-22.

Nye, J. F., and F. C. Frank. "Hydrology of the intergranular veins in a temperate glacier." Symposium on the Hydrology of Glaciers. Vol. 95. Cambridge England, 1973.

Raymond, C. F., and W. D. Harrison. "Some observations on the behavior of the liquid and gas phases in temperate glacier ice." Journal of Glaciology 14.71 (1975): 213-233.

Rempel, A. W., et al. "Possible displacement of the climate signal in ancient ice by premelting and anomalous diffusion." Nature 411.6837 (2001): 568-571.

Thorsteinsson, Thorsteinn, et al. "Crystal size variations in Eemian-age ice from the GRIP ice core, central Greenland." Earth and Planetary Science Letters 131.3-4 (1995): 381-394.
* * *

---

## Author Comment (AC1) · 24 Oct 2020

Initial response to Reviewer 1

Thank you for your appraisals and careful and detailed review of the manuscript, and for providing a range of valuable suggestions for improving it, including phrases that can be used directly in the revision.

In the revision, I plan to show some examples of ice core records in a new figure, probably early on in the manuscript. As you pointed out, this will help non-specialist readers. Besides showing the length scale and typical character of the peak signals, the figure will help motivate the choice of the doped peak amplitude used in the numerical experiments. Probably I won't comment on the variability and trends observed

on the records, as these may generally reflect a combination of palaeoclimatic and post-deposition changes. Also I wish to avoid giving readers the misimpression that the simulated decay of cB in the manuscript necessarily explains the reduction of variability with depth that has been pointed out for some of the records. This is because cB in the model refers only to the vein component of the bulk ionic concentration, whereas the bulk concentration in the ice core records is the total concentration (potentially) consisting of vein, grain boundary and matrix contributions.

Many of your other suggestions will be taken up. I will adjust the caption of Figure 1 to explain how the schematic relates to position in the ice sheet. I will add a list of model variables, although I have not decided whether this is best done by a separate table or by extending Table 1. There are different perspectives regarding the choice of "we" versus "I"; my preference is to use "we" in this manuscript. Where cB = c*Phi is introduced, I will clarify that this equation is valid because cB in the model refers to the vein component of the bulk impurity concentration, and excludes contributions from microinclusions and grain boundaries.

I appreciate your comment about the age span and it is useful to know how much this parameter interests ice core scientists. I will try to write more about this parameter, by highlighting some of the ideas that you expressed in your review. The discussion is going to be brief, limited to a short passage, because the model runs assume steady-state forcing for the age depth scale and does not tackle the inverse problem of calculating the ice flow with varying accumulation rate and temperature histories (see lines 276-278), and so the deep results are also strongly dependent on the assumed approximate age-depth scale (lines 363-364) and can only provide a rough indication of the age span. To put this another way, the results apply strictly to the "model ice core" and not necessarily to the actual data in the EPICA Dome C core. Before the time-dependent problem has been solved, it is best to be cautious and not to interpret too much into what the specific age span values (especially at depth) in Figures 6 and 7 imply for the climate histories retrieved from the cores. In the passage to be added, I

will try to mention these caveats as well as the insights.

It is possible to rescale the age span axis in Fig. 7f to show the behaviour of the black and grey curves at depth. I have decided not to do this, not only for the reason given in the last paragraph. In the simulation runs yielding the black and grey curves, the signal amplitudes have actually diminished to almost zero at t > 400 ka (Fig. 7d). At such large ages, only those age span results for the runs with diffusion suppressed (the blue and orange curves) are worth showing, because there is still some signal amplitude, and Fig. 7f is currently scaled to portray those curves.

A full response addressing your comments point-by-point will be given later during the revision stage.

––––––––––––––––––––––––––––––––

---

## Author Comment (AC2) · 26 Oct 2020

Initial response to Reviewer 2

Thank you for your positive and insightful review of the manuscript, and for providing a useful and engaging summary of its findings.

Your main suggestion is for the manuscript to clarify the different assumptions used in the present model and in the original model of Rempel et al. (2001), regarding the possible effect of grain size variations on the Gibbs-Thomson term in equations (7) and (8), and notably the idea that inverse coupling between grain size $d_g$ and bulk impurity concentration $c_B$ could make the vein face radii nearly uniform (causing the Gibbs-Thomson term to become constant) and prevent diffusion from occurring. I am happy to

discuss this matter in the revision, and agree that such discussion can stimulate further study of the mechanisms behind grain-size variations. I plan to insert the passage and the associated discussion in the final section of the manuscript, and only briefly signpost the matter earlier, as the corresponding arguments are involved and contain intricacies that will be difficult for readers to appreciate and will disrupt the thread of the work if this discussion is delivered in the earlier sections.

I have a few reservations about the model assumption stated by you in the review for the Rempel et al. (2001) study. I think that those reservations need to be included in the new passage, in order to keep the reader informed of different viewpoints. They are given at the end.

Before describing them, I have a request. In the review, you wrote:

"... the original reasoning provided by Rempel et al. (2001, p. 570) emerges from the assumption that an anti-correlation between grain size and impurity loading makes vein radii uniform, as predicted if grain size scales inversely with the square of bulk impurity loading in equation (7) of the current work, thereby rendering the second term in equation (8) spatially uniform as well."

This is an important clarification, as I don't think it is explicitly clear from the writing in their paper that their model had ignored the Gibbs-Thomson effect for the reason given above. The final sentences of their first paragraph on p. 570 do not relate the vein radii, via vein face curvature/radius, to the Gibbs-Thomson effect nor to depression of the melting temperature. Therefore I think that issuing this clarification will be useful to the field; conveniently, all review materials in The Cryosphere are also citable so that readers can trace its origin to this review discussion. Both the physical concept being stated, and the clarification of its role as an assumption in the Rempel et al. (2001) model, should be attributed to the originator rather than me. For this reason, can you please give me permission to reference the stated ideas to you via "personal communication"?
The assumption that you have given for the Rempel et al. (2001) model goes as follows. It is posited that grain recrystallisation processes (at least, extraneous processes outside the formulation in section 2.1) lead to $dg^2 cB$ = constant, where $dg$ is the mean grain size and $cB$ is the bulk ionic impurity concentration of the vein network. Then, the second part of equation (7) yields a constant vein face radius $rv$, and the Gibbs-Thomson terms in equations (7) and (8) become constant at a given temperature (here I add that the same would be true for the general form described on lines 132-134 of my manuscript). The Gibbs-Thomson diffusion thus vanishes. Support is provided for the assumption, from the observed anti-correlation between grain size and impurity loading that has been reported by numerous ice core studies.

Here are my thoughts regarding this assumption:

1. The desired anti-correlation between $dg$ and $cB$ needs to obey $dg^2 cB$ = constant exactly, for the diffusion to vanish. Moreover, an anti-correlation may not necessarily alleviate the diffusion: it could enhance the diffusion. For example, suppose that recrystallisation processes cause $dg \propto 1/cB$. Equation (7) then yields $rv \approx \sqrt{1/cB}$, rather than $rv \approx \sqrt{cB}$ (as indicated currently by the equation when the grain size varies slowly). Note that the dependence of $rv$ on $cB$ has not gone away. With the anti-correlation, the Gibbs-Thomsom term in equation (8) now goes as $cB^{0.5}$, instead of $cB^{-0.5}$. This merely gives rise to a different nonlinear diffusion term on the right-hand side of equation (23). For certain values of the coefficient of proportionality in the relationship $dg \propto 1/cB$, the diffusion can actually be stronger than currently predicted in my simulations (and this is true for many other kinds of negative relationships between $dg$ and $cB$). Consequently, anti-correlation between $dg$ and $cB$ does not generally support the model assumption that would prevent the Gibbs-Thomson diffusion from operating — it may support the opposite.

2. Although my numerical experiments prescribes smoothly-varying grain size profiles, the formulation in section 2.1 (equations (1) to (8)) remains general, in the sense that coupling of the grain size $dg$ to other variables via recrystallisation processes is allowed (as indicated on line 237). I would hesitate to describe equations (1) to (8) as involving assumptions that render the present model as an "end member case". This is especially because it is possible for inverse coupling of dg to fluctuations in cB at short length scales to cause or even enhance the diffusion of signals (see Item 1). In contrast, I think that the specific assumption of dgˆ2 cB = constant comes across as a special case in the application of the model in equations (1) to (8).

3. The bulk impurity loading shown by ice core studies to exhibit anti-correlation with the grain size is the sum of impurity contributions from the ice matrix, grain boundaries, and the vein system. However, the model variable cB, which needs to satisfy dgˆ2 cB = constant for the desired assumption to hold, refers to the ionic impurities in the vein network only. Accordingly, arguments that use the observed anti-correlation to support the assumption need to explain how the vein impurity loading relates to the total impurity loading; this explanation is missing. (As discussed in the manuscript, these quantities are not necessarily proportional to each other.)

4. It is not clear to me what physical mechanisms would enable the concentration of impurities in the veins (which make up cB) — as opposed to impurities at grain boundaries and in the ice matrix — to control the grain size. Existing theories that relate grain size variations to ionic impurity loading consider how dissolved impurities can reduce grain boundary mobility, through the production of drag force on grain boundaries (e.g. Alley et al., 1986, p. 422), but such mechanism refers to the impurities situated at grain boundaries, not to impurities in the vein network located at three-grain junctions.

The ideas in Items 3 and 4 indicate that for the desired assumption to hold, either some unknown/unexplained mechanism exists to allow the impurity in the veins to influence grain-scale recrystallisation processes in such a way for cB to control the mean grain size in the right manner, or grain-boundary (maybe also matrix) impurities control the grain size dg and separately regulate the vein component cB in just the right ways, to satisfy dgˆ2 cB = constant. I think that one has to be quite hopeful for either set of interactions to yield the right behaviour to justify the assumption, especially as impurity

concentration is only one of several factors known to affect the mean grain size. Having said this, I have not explored these theoretical possibilities (the present manuscript is not about how recrystallisation processes and impurity factors control grain size variations) so I do not rule them out, and they can be investigated in future research.

As mentioned before, I am happy to add a passage that covers your clarification of the assumption behind the Rempel et al. model, how it relates to the formulation in the present manuscript, and also the above elements.

Reference: Alley, R. B., Perepezko, J. H., and Bentley, C. R.: Grain growth in polar ice: I. Theory, J. Glaciol., 32, 415–424, 1986.

---

## Referee Comment (RC3) · Alan Rempel (Referee) · 29 Oct 2020

I agree that it would be helpful to signpost a later discussion on the importance of grain-size variations for determining how vein constituents behave. I would also welcome having the ideas surrounding the specific assumption regarding uniformity of vein radii in the Rempel et al. (2001) model attributed to a personal communication. However, I do not think that such a reference is necessary. While it is true that the matter was not discussed at length in that work, the reasoning was explicitly provided by the statement that "as the surface energy of curved interfaces acts to make vein radii uniform, variations in cB must correlate with changes in the total length of veins per unit sample volume", with the following sentence going on to note the qualitative support provided by observed anti-correlations between bulk impurity content (cB) and

grain size.

The twenty years since we completed that work have seen tremendous advances in the community's ability to characterize the physical and chemical characteristics of ice cores at increasingly fine scales. Despite these advances, it is noteworthy that we still lack a quantitative mechanistic understanding for precisely how the anti-correlation that is commonly observed between cB and grain size develops. Your new model brings welcome attention to the consequences of that important issue. In your comment you: 1. highlight the importance of the detailed form of the anticorrelation for determining the fate of vein constituents, 2. champion the generality of your formulation in being readily adaptable to examine the affects of different grain size evolution laws, 3. assert the need for a mechanistic understanding of the role of impurity loading on grain size, and 4. emphasize the inability of existing theories of grain growth to address this problem. I think we have broad agreement on each of these points, which offer a clear motivation for filling these knowledge gaps and bolstering confidence in the integrity and resolution of these important paleoclimate records.

Beyond the observed fine-scale grain-size variations themselves, an argument in favor of the uniform vein radius assumption employed in the Rempel et al. (2001) treatment is the long-term preservation and apparent fidelity of fine-scale cB signals recovered from ancient ice. If, as in your model treatment, vein radius evolves to force diffusive impurity redistribution, then your analysis implies that either those deep signals are distorted from their original form, thereby compromising detailed paleoclimate interpretations, or instead their preservation might be attributed to one or both of the mechanisms that you suggest, namely: residence outside of the vein network under much warmer conditions than the eutectic temperatures of their solutions, or blockages that manage somehow to severely restrict vein diffusivity. The observed anti-correlation between impurity content and grain size would in any case remain unexplained. However, should this problem be addressed, the precise manner in which grain sizes respond to impurity content or perhaps some coincident variable (e.g. impurities on two-grain boundaries)

could be accounted for in a refined treatment that extends beyond the locally uniform grain size case that is the focus of the example calculations in your paper. While we're each free to argue over the set of assumptions we feel to be most reasonable, whichever situation actually dominates is not currently known.
* * *

---

## Author Comment (AC3) · 29 Nov 2020

———————————————-

Final Author Response to all reviews

———————————————-

Reviewer 1 gave an engaging summary and detailed appraisal of the manuscript. Two key points are that (i) its appeal and accessibility to non-specialists would be improved by adding a figure to exemplify ice-core ionic records, and (ii) the simulated age spans of the signal peaks in Figure 7 can be further discussed, in relation to the observed signal variability deep in the EPICA Dome C core. I plan to implement both recommendations in the revision. The added figure will probably show two records (including one

for sulphate), with the aim of illustrating the abundance of the peak signals, their form and length scales, and their presence in deep ice; the records won't be examined beyond this, as detailed analysis of them is best left to the original papers. Regarding (ii), I will try to discuss the age span results a little more along the lines suggested by the reviewer; this will be brief, as how much matrix/grain-boundary impurities contribute to the observed variability remains unknown, and some of the apparent variability may derive from the merging of signals (Section 3.3) rather than from single peaks. In the minor comments, Reviewer 1 also provided valuable suggestions of local wording for different parts of the manuscript, and quite a few of them (especially those that improve writing clarity) will be taken up. For my earlier detailed response to Reviewer 1, please see AC1.

———————

Reviewer 2 provided a helpful summary of the manuscript's contribution and a focussed suggestion that during the revision, I should add discussion to contrast the present model against the original model of Rempel et al. (2001), in regard to the possible effect of grain-size variations on the Gibbs-Thomson term in equations (7) and (8), and the consequential effect on signal diffusion. As I responded before in AC2, I am happy to do this, especially as the discussion may invite more research to examine the mechanisms of grain-size evolution.

Reviewer 2 clarified that Rempel el al. (2001) had ignored the Gibbs-Thomson effect by assuming that inverse coupling between grain size and bulk impurity concentration makes the vein face radii uniform, and that those authors invoked the observed anti-correlation between grain size and measured ionic concentrations in ice-core records as evidence to support this assumption. I don't think that the writing in Rempel el al. (2001) conveyed this assumption clearly/completely (that the Gibbs-Thomson term was neglected on the basis of uniform vein radii was not said; and there are more issues about the support for the assumption, as highlighted below), and I find it intriguing that both follow-on papers by Rempel et al. (2002) and Rempel and Wettlaufer (2003)

(exploring the ramifications of Rempel theory) had not once referred to that assumption, but instead elaborated upon an entirely different assumption in order to neglect the Gibbs-Thomson term – via its small size compared to other terms. In any case, I am happy to relate this clarification as made by Reviewer 2 by referring to RC2.

My plan is to add the passage discussing grain-size variations to Section 4, using the above as one of the starting ingredients. It will be noted that adding grain-size variations will modify the rate of signal diffusion in the revised theory, and that the model formulated by me in the manuscript (Section 2) allows any grain-size prescription, even though the numerical experiments in Section 3 employed smooth grain-size profiles, for reasons given on Lines 237-240. Then, the possibility that Gibbs-Thomson diffusion would vanish identically *if* grain-size variations (at short length scales) occur in such a way that the bulk vein impurity concentration $c_B$ and mean grain size $d_g$ are coupled via the highly-specific relation $c_B \propto d_g^{-2}$ will be discussed. This scenario, favoured by Reviewer 2, is what he clarified to be the assumption behind the Rempel et al. (2001) theory. I think its consideration enriches the manuscript by furthering our query of under what circumstances might signals be able to escape diffusion to show migration at depth. However, there are fundamental issues with arguing for this scenario by using the observed anticorrelation between grain size and impurity loading in ice cores; to state them briefly: (i) Inverse coupling between $c_B$ and $d_g$ does not generally imply suppression of the Gibbs-Thomson diffusion. Depending on the form of the inverse relation, it can enhance or weaken the diffusion (see my writing in AC2 for details); only strict obeying of $c_B \propto d_g^{-2}$ would suppress the diffusion, so this condition is highly specific/restrictive. (ii) In the assumed scenario, it is the bulk *vein* impurity concentration $c_B$ that must obey the inverse-square relation with $d_g$. However, the evidence used for support pertains not to $c_B$; instead, the observed anticorrelation concerns the bulk impurity loading measured in ice cores. One cannot simply equate this loading to the vein concentration $c_B$, because the loading contains an unknown contribution of matrix and grain-boundary impurities. (iii) No physical mechanism has been offered for why increasing $c_B$ (in the veins) would reduce the grain size; existing

theories on how impurities impede grain growth consider the effect of impurities at grain boundaries only, and not impurities in the veins. [See AC2 for further details on (i)-(iii).] These issues, which will be described in the new passage, mean that convincing physical arguments and evidence for this special conjecture to hold ("zero Gibbs-Thomson diffusion because cB \propto dgˆ{-2} is satisfied throughout the ice column") are currently lacking.

From these, the reader can decide whether to regard the original theory (no diffusion, only migration) as describing the predominant behaviour of vein ionic signals in ice sheets. In contrast, in the manuscript I did not claim that the signal diffusion rate being simulated is precisely accurate nor that grain size profiles are always smooth. The point is to show what happens to signals when the Gibb-Thomson term is there – something which had been overlooked.

---

## Author Comment (AC4) · 1 Dec 2020

Final Author Response to all reviews [postcript]
* * *
Based on helpful remarks by Reviewer 2, I wish to issue two clarifications of my writing in AC3:

1. In Paragraph 4, I wrote: "I find it intriguing that both follow-on papers by Rempel et al. (2002) and Rempel and Wettlaufer (2003) (exploring the ramifications of Rempel theory) \*\*had not once referred to that assumption\*\*, but instead elaborated upon an entirely different assumption in order to neglect the Gibbs-Thomson term – via its small size compared to other terms."

The ** phrase is to be taken to mean "had not once referred to that assumption FOR THE PURPOSE OF NEGLECTING THE GIBBS-THOMSON EFFECT". This meaning should already be clear because of what is written in the following phrase, because of the paragraph's context, and because Lines 1-2 and Lines 6-7 of the paragraph identified the assumption as that particular assumption used to ignore the Gibbs-Thomson effect. I offer this note so the meaning cannot be mistaken.

2. Editing change to Paragraph 5, Line 10: Replace: "favoured by Reviewer 2" by "pointed out by Reviewer 2". I apologise to Reviewer 2 if the original wording did not capture his relation with the idea correctly.

---

## Author Response (AR2)

23 January 2021

Dear Editor:

I have undertaken a thorough minor revision of the manuscript, addressing all of the reviewers' points, following many of their suggestions. I have done more, by adding a substantial Section 3.4 to provide analysis and simulations to show how grain-size fluctuations could *create* prominent, *non-migrating* impurity signals. This section not only covers a topic that interests Reviewer 2, but also goes much further than that. The findings of my study of are thus enriched; its core arguments and conclusions are unchanged. Relevant parts of the Abstract, Introduction (Sect. 1) and Conclusions (Sect. 4) have been adjusted to coordinate with this addition.

A new Fig. 1 (examples of ice-core records) and Table 2 (mathematical symbols) have been added, following Reviewer 1's suggestions. The original Figs. 1 to 10 are renumbered 2 to 11.

Fig. 12, Fig. S3, Movies S6 and S7 have been added to accompany the new Section 3.4 reporting the findings about the potential impact of grain-size fluctuations.

Movie S8 has been added to illustrate how a signal in an ice core during cold-room storage would change in time. It is used by the last Conclusion paragraph in Sect. 4.

The Supplementary File and the Data Repository have been updated for the new figure and movies. See: https://figshare.com/s/8607e837455c5188c207

My detailed response and description of changes are written below. The reviewers' comments (from RC1, RC2, RC3) are shown in blue.

On page 3 of the revised manuscript, you will see a footnote, referred to on Line 67. I read the TC guidelines: "*Footnotes should be avoided in the text, as they tend to disrupt the flow of the text. If absolutely necessary, they should be numbered consecutively.*" In this instance, I believe that the footnote is an absolutely-needed case as its content would disrupt the main text if embedded in it (if not as a footnote). The information is peripheral but of interest to specific readers, and I think that keeping it as a footnote allows the main text to flow well.

Following TC instructions, a marked-up PDF file showing all changes (in MS-Word track change) is attached to the end of this response file.

Thank you for your attention.

Best wishes,
Felix Ng
* * *
Detailed response and description

L = line number in revised manuscript

**RC1: Anonymous Referee #1**

The paper by Ng tackles an issue from a theoretical point of view that bothered ice core
science for nearly two decades, i.e., the potential signal migration of bulk chemical
signals through the vein network in the presence of an in situ temperature gradient.
This hypothesis was put forward by A. Rempel in 2001, but while no unambiguous ice
core evidence for this hypothesis was provided in the meantime it could also not be
refuted as the theory of migration through the vein network is physically not disputed.
However it is still unknown how much of the impurities are located in veins and, thus,
really subject to this transport.

The results by Ng provide a convincing argument why no evidence for a migrated signal
CAN be found, i.e., because any migration signal would also be subject to intense diffusional
smoothing. Ng expands on the theory by Rempel by taking into account the curvature effect on vein
volume that had been intentionally neglected by Rempel. Ng is able to convincingly show that taking
this additional term into account, does not stop any migration but leads to strong diffusion of such a
migrating signal and, thus,

to the disappearance of the displaced signal in the ice. This has two very important
implications for ice core science: Firstly, any short-lived peaks in the ice core record
that are imprinted in the veins cannot survive the migration process. Vice versa, if
there is a peak found in the ice core record it cannot be shifted relative to its initial
bulk position. Secondly, the presence of distinct peaks in many deep ice core records
suggests that vein migration and diffusion cannot be a dominant process as otherwise
any peaks would have disappeared. This implies, as explicitly outlined in the paper by
Ng, that either the part of impurities located in veins is only small or that vein transport
is increasingly suppressed by a fragmenting vein network as already suggested by other authors.

The paper is well written and excels by its stringent, mathematical approach, while at
the same time performing model experiments that are very instructive and, therefore, of
great value for the ice core practitioners. It is therefore of high relevance and well suited
for publication in The Cryosphere. The mathematical approach (although well laid out)
and some for the outsider unintuitive formulations make the access to the paper a little
more difficult than necessary. Below I make some suggestions for minor revisions that
may help to remedy that. In summary, I highly recommend the paper for publication in
The Cryosphere after taking care of these minor revisions.

I thank the reviewer for the appraisals and the valuable suggestions below.

**General comments**

The paper is a little "dry" as there are no ice core examples provided. Thus, for those
readers who are not specialists in ice core chemistry, it is hard to imagine how those
signal that Ng talks about, look like. Here, it would be helpful to see (in the Introduction
or the Discussion) a figure that shows examples for high peaks in deep ice on the
one hand (for example for in the $Ca^{2+}$ or $SO_4^{2-}$ record) and examples of a general
reduction of variability as seen on others (for example in the $Na^+$ and $NO_3^-$ record).
Examples can be found in the papers by (Traversi et al., ES&T 2009, Röthlisberger et al., CP 2008,
Schüpbach et al., Nature Communications 2018, Barnes et al., JGR 2003, and others).

The new Figure 1 has been added to show ice core examples, from Antarctica and Greenland. Thank
you – this addition really enriches and embellishes the paper.

As mentioned in the interactive discussions, I do not evaluate the variability/trends observed on such
records, and leave that to the observational / future studies. The focus is to illustrate the abundance of
ionic peaks and their expressions and variety in deep ice. Thanks for pointing me to Schüpbach et al.
(panel c). Also, R. Traversi kindly provided data to me, allowing me to make panels a and b. And I
managed to find a published piece of high-resolution data for NGRIP (panel d). Other than this, I have
not found usable data, as publicly available/archived chemical data at a resolution of 50 cm or better
are scarce. A resolution as high as 10 cm is necessary to portray the peak forms for my purpose.

L53 and L331 now refer to Figure 1.

In addition to Table 1, which provides the values of the constants used, I would suggest to provide a table 2 where all variables used are listed with a short explanation and their units.
Done. Table 2 has been added. To avoid a very long table, I restrict it to the model variables appearing in the main text, and exclude those variables local to Appendices A and B.

The information on the age span is provided in figures 6 and 7, however the discussion of this parameter, despite its great importance for ice core science in particular for very old ice, is rather limited. Looking at the EPICA Dome C results it appears that the age span in Fig. 7 approaches several thousand years or even the precession age scale when reaching an age of 700-800 kyr (unfortunately the current scale of the x-axis does not allow to quantify this for the control run). On the other hand, the EPICA Dome C ice core record provides some information about the time scale of variability that can still be resolved, thus, would constrain the degree of vein transport smoothing empirically. I would recommend to extend the discussion on this point, as it is of great relevance for ice core sciences.
Age span (revised paragraph on L366 to L380):
Discussion has been added on L370–373. I appreciate your interest in seeing the computed age spans being used to interpret the actual details of the EPICA ice core at depth. However, I caution against doing so, giving a specific reason on L372-373, and adding another overarching reason on L377–380. The opening sentence of the paragraph (L366) now includes "age span" as a signpost to the paragraph's topic. Note that the old Figs. 6 and 7 are now Figs. 7 and 8.

The horizontal "age span" axes in Fig. 8c and 8f are not scaled for the control runs (their axes focus on the other runs) because in deep ice, the signal amplitudes have diminished to near zero, as shown in Figs. 8a & 8d. I have added a final sentence in the caption of Fig. 8 to explain this (L1005–6).

**Specific comments:**

Introduction 1st paragraph. You could also mention here already the loss of species such as NO3- or Cl- at the surface (Röthlisberger et al., Ann Glac 2002, Weller et al., JGR 2004) migration processes (for example for methanesulphonate) that occur already at the surface (Osman et al., 2017) or the aggregation of dust particles in the deepest ice (Tison et al., 2015)
I have not done this. I prefer a more focussed thread leading into the core matter of the study (2nd paragraph onward) and feel that adding these details to the 1st paragraph would detract from that.

p2 l33: "...migrate relative to the ice..."
Done; L33.

p2 l35: "... could decouple..."
Done; L36.

p2 l41: "... signal migration in deep ice may..."
On L42, I have not followed your suggestion to write "in deep ice", because signal migration (the process) can occur --- and be limited --- at any depth. (It is true that large displacements manifest at depth as a result of signal migration.) I therefore prefer the existing wording.

p2 l51: this is one of the examples where the author refers to the ice core evidence without showing it
Thank you. The new Figure 1 showing ice-core records is now referred to on L53 in a new sentence. I adjusted the wording on L52 to coordinate with this.

p2 l62: One intrinsic assumption made in the Rempel theory and also in the work by Ng is that $c_B=c*Phi$. However, if impurities are located in microinclusions or grain boundaries, hence not in contact with the veins, then the vein transport is not representative of changes in bulk concentrations. In fact, this is discussed later, but I would recommend that this assumption is explicitly mentioned when $c_B=c*Phi$ is introduced.

Yes, I now mention this assumption explicitly by:
(i) clarifying the definition of $c_B$ (including its units) on L58–60, and
(ii) writing "With $c_B$ encapsulating vein impurities, the relation $c_B = c*phi$ holds…" on L66–67.
Alongside these changes, an opening phrase on L105–106 has been modified.

In the caption of the New Figure 1, I also clarify that the impurity concentration of an ice-core record is the total concentration, whereas $c_B$ refers only to the vein impurity component (L868–869).

p3 l63: when I first read the word "ice porosity" I got confused. I would suggest to write:
" is mirrored by variations in the liquid filled vein volume relative to the total ice volume, in the following called ice porosity Phi"
On L66, I now write "porosity" instead of "ice porosity", and I define this term in the same sentence by writing
"… in the porosity (Fig. 2c, d), which represents the volume fraction of veins in the ice."

p3 line 68: please change the unclear wording "Thus the c_B peak translates"
Done. Changed to "Thus the peak signal in $c_B$ translates"; L71. (On L71–72, in order to be informative, I added a phrase in brackets to say that "the same translation applies to trough signals".)

p4 l96: replace "we" by "I" throughout the manuscript
I prefer not to do this in this manuscript. I understand that some scholars prefer what you suggest.

p4 l111: "The three terms on its right-hand side describe the temperature depressions due to (i) solute, (ii) interfacial curvature (the Gibbs-Thomson effect) and (iii) pressure, respectively;"
Done; L114–115.

p4 l 113: what does the constant gamma represent?
Now clarified on L116 (gamma is interfacial energy). I adjusted the entry for gamma in Table 1 to coordinate with this change.

p6 l147: "... where the melt rate m (in units?) at the interfacial boundaries of the vein accounts..."
Done. On L153–154, I have clarified these in two linked sentences.

p6 line 155: why not explain the last term here?
Revised. In the old paragraph, the reader was meant to understand that the last term describes vein motion (which was treated previously, for porosity), but the paragraph's structure did not pull this off well enough. I have now rewritten the first 3 lines of the paragraph so that which term corresponds to which process is clear. Please see L160–162. I also remind the reader that the vein motion is the same vein motion that was considered a few paragraphs ago (L161, phrase in brackets).

p7 equations 17 and 18: You neglect dw/dz. Say explicitly why.
Done; L178. I say here that incompressibility (Div.u = 0) has been used in deriving this result (dw/dz wasn't neglected; dw/dz + du/dx + dv/dy sums to zero). This clarification on L178 is paired with an earlier clarification on L170.

p11 l256: "while anomalous (Rempel) diffusion"
Done; L265.

p11 l273-274: use a different variable for the basal melt rate (for example m_bas) to distinguish it from the interfacial melt rate m used above
Done; L283–284.

p12 l298: Likely a topic to be looked at in a separate paper, but the theory by Ng should be also able to predict how much vein diffusional smoothing occurs for the water isotopes.
Thanks for this suggestion. It is best left to a separate paper, and I haven't modified the text, because while one can consider diffusion of (i) signals of water-vein ionic species and (ii) signals of isotope abundance, some key physics differs between (i) and (ii). Topic (i) involves the solute-controlled liquidus, while (ii) doesn't. Topic (ii) involves equilibrium fractionation but (i) doesn't. There are broad parallels but I expect the model equations to be rather different.

Thanks; done; L328.

Thanks for pointing this out. On L331–332, the Mayewski references have been removed. I have
added your suggested references (except Bigler et al., whose study extends to 2 km depth only and
thus somewhat marginally illustrates my sentence) and included a reference to Svensson et al.
(2013), whose data feature in Fig. 1d.  Also, on L331, I refer to the New Figure 1.

Done; L342.

Done. Please see my earlier detailed response to your last *General Point* above. The revised
paragraph is on L366 to L380.

Great to know!

Done; L388–389. Here I clarify that the movement is due to vertical ice compression.

I haven't done this for the movies (e.g. Movie S3) because the temperature curve would still jump
from frame to frame, as MATLAB (my plot-making software) puts its 'tickmarks' to span the y-axis so
that the top and bottom tickmarks always lie at the corners of the plot. I am not enough of a "Matlab
guru" to know how to control the plot element properties to be able to overcome this.

Done. Please see L419–421. For the clarification part, I have rewritten the Original L403–404 as
        "it is understood that fewer high-frequency palaeoclimatic details are retrievable from deeper
ice, due to the finite resolution of ice-core sampling, alongside layer thinning, which causes more time
to be encapsulated in a given ice thickness."
I have not discussed the time scale that could be resolved, because of my reservations (expressed
before) about using the model-computed age spans to analyse the details of real ice-cored records, in
deep ice (see my earlier responses on "age spans"). The analysis of real records falls outside the
scope of the present study and I feel that this is best left to another piece of work.

Yes. In response to your earlier point for "p2 l62", the assumption behind $c_B$ = c*Phi and an explicit
definition of $c_B$ have now been given early in the manuscript.  Also, I have now made several minor
wording changes in Section 4 so that whenever the symbol $c_B$ is used, it refers only to the vein
impurity component, whereas the *measured* impurity records quantify the total concentration ---
including vein, grain-boundary and matrix contributions. The corresponding changes can be found on
L495-6 (bracket), L511, and L532.

Reworded now as "the signals in $c_B$"; see L543.

Yes, on L548–550, I have now removed the reference "… *as done in volcanic flux reconstructions
from SO42-*" and rewritten the sentence – in order to be specific – as:

"These considerations caution against interpreting all observed ionic signals directly for palaeoclimatic events and variations: some signals may be distorted in form and duration, and some peaks may be caused by local grain fining (this may result from recrystallisation processes (Faria et al., 2013) or high levels of dust/microparticles in the ice (Alley et al., 1986a))."

As explained above, it is beyond this paper's scope to address the specific details of measured records with the computed age spans.

p18 l2: "of any vein impurities"
Done. Replaced "all" by "any"; L554.

p18 line 476-478. expand the discussion on longer variations
I have deleted this short paragraph from the middle of Section 4. The longer variations had already been discussed in the final paragraph of Sect. 3.3 (L414–423) and I have no more to add here.

p19 l498-500: This comes unexpected and justifies a little bit more discussion.
Yes. The paragraph at the end of Sect. 4 was a little abrupt: its brevity didn't help. To inform readers more, I have extended it by describing how fast a signal in $c_B$ would diffuse/decay at two storage temperatures (see L591-596; L593–596 are new). Movie S8 is added to show the simulated results.

Figure 1: I know these are only illustrative figures, but the scale of the anomaly (centimeters) and the approximate location within the ice sheet of the sketch (lower third, where temperature gradients exist both in Greenland and Antarctica) should be indicated either in the figure itself or at least in the caption.
Done (note that Figure 1 has been renumbered as Figure 2). I followed your second suggested option. To the caption of Fig. 2, I added a sentence to describe the whereabouts of these gradients in an ice sheet, referring also to the actual examples shown in Fig. 4; see L889–890. Now L890-891 describes the scale of the signal/anomaly in panels b, c and d, without pinpointing where though, because such signals can occur anywhere in the ice column.

Figure 2: clarify what you mean by "that measures distance from ice at age t"
Done. I now clarify this idea in two places:
- caption of Fig. 3 (Old Fig. 2), L902–905 have been expanded to clarify the idea;
- L252–253; rephrasing done here to elaborate the idea.

Figure 7: rescale panel f to show the age span for ice with an age of 800 kyr.
[Note: this figure is now Figure 8.] As explained earlier in my response, I do not rescale the age-span axes here, because the signals in the control run have decayed to near-zero amplitude at depth, but on L1005–1006 I have added a final caption sentence to explain the choice of axis scaling.

Again, I thank this reviewer for his appraisals and suggestions.

Rather than to recount my interactions with Reviewer 2 during the *Interactive Discussions* process, here I summarise all of what I have done to manuscript following those interactions.

Those interactions, while valuable, have been mostly confined to one idea raised by the reviewer (and the surrounding considerations) --- his idea of how the Rempel theory might be justified under specific conditions. I think that this topic is useful, but has a limited value for the manuscript and for readers.

I went much further than covering that topic. In a New Section 3.4, on page 16–18, I analyse the model more and report what happens to cB if we *prescribe* a grain-size fluctuation in the ice. That is, fluctuation in $d_g$. This includes a short piece of mathematics (L455–466) to explain the interactions, and a set of simulations showing the results (L467–488). The fact that these preliminary/artificial experiments do not involve a physically-based model of grain-size evolution (because this is out of reach) is again made clear (L425–430, L471–472).

The discovery is that a grain-size fluctuation can cause a new signal in cB to form, and that *the resulting signal is locked to the fluctuation and does not migrate relative to the ice*. Section 3.4 is accompanied by the addition of Fig. 12, Fig. S3 and Movies S6 and S7. These results extend the paper's findings. To coordinate with them, I have adjusted various wording and inserted qualification/signposts elsewhere in the manuscript:

- Abstract (L15–16, 18–20, 21–22 & 23, reporting and qualifying findings in regard to $d_g$ fluctuations)

- Introduction (L55, signposting the new result)

- Model section (L135–7, L215–6, L248–9, sentences added to signpost later work with $d_g$)

- Discussions (L495–500, qualification added "*unrelated to grain-size fluctuations*")

     (L514, 520–521, 527–531, 536, 546–547, 549–550, 552–553, 560, 569–570, 583, 589, embedding of the new results concerning $d_g$ and their ramifications).

Early within the New Section 3.4, I also carried out my plan given in AC2 and AC3. On L430–L440, I added a paragraph to describe the concept --- raised by Reviewer 2 in RC2 and RC3 --- that a specific inverse-square coupling exists between mean grain size $d_g$ and vein impurity concentration $c_B$ to keep vein radii $r_v$ uniform, so that the Gibbs-Thomson term is kept uniform and this suppress the Gibbs-Thomson diffusion. In the next paragraph, on L441–L454, I evaluate this idea, noting my reservations of it by considering its mathematical, empirical and theoretical bases. I think that these two paragraphs of 500 words suffice to address this topic which interests Reviewer 2.

Regarding Reviewer 2's idea that the observed anticorrelation between *total* impurity loading and grain size ($d_g$) in ice cores justifies the necessary coupling between $c_B$ and $d_g$, I remain unsure. This is because the *total* measured loading (for an ion species) generally consists of 3 contributions: (i) impurities in veins, $c_B$, (ii) impurities at grain-boundaries, and (iii) impurities in the ice matrix/crystals, e.g. in microinclusions. Suppose we have measurements of a total quantity T = X + Y + Z, but have no measurements of X, Y, Z. The observation that "*T is inversely correlated with another variable P*" does not mean that "*X is inversely correlated with P*". Much as I recognise such statement as a possibility, it is not supported. With the currently available evidence, this is as far as I can go in terms of evaluating this particular idea --- please see L445–448.

Below, I copy and paste all of the text of RC2 and RC3, but have not written point-by-point response against it, as my overall response is given here.

**RC2:**

The manuscript by Felix Ng revisits the question of what happens to soluble impurities
in glacial ice over the long time periods represented by ice-core climate records.
Assuming that these impurities are primarily contained within a connected liquid-filled

vein-node network along three- and four-grain contacts (e.g. Nye and Frank, 1973), an earlier model predicted that the control of impurity loading on liquid content should produce spatial variations in the effective compositional diffusivity that would help to explain the long-term preservation of short wavelength (e.g. representing seasonal to decadal time periods) bulk compositional signals, while also leading to their translation relative to the surrounding ice (Rempel et al., 2001). This so-called "anomalous diffusion" phenomenon has proved very difficult to test, since the predicted rate of signal migration is very low (largely controlled by the shallow temperature gradients in polar ice) and firm constraints are rarely, if ever, available on the relative timing of compositional and ice-borne (e.g. oxygen isotopes) proxy deposition in the distant past. If the compositional signals themselves aren't altered in form, but only displaced a small distance relative to the ice with which they were originally deposited, what evidence is there to unequivocally demonstrate such subtle effects?

Ng's analysis predicts qualitatively different behavior in a slightly modified system, by focussing on the role of vein surface energy embodied in the Gibbs-Thomson effect, while assuming that the ice is characterized by locally uniform (but slowly growing) grain sizes. The models of anomalous diffusion that had been presented earlier had differed by employing a different assumption concerning the relationship between ice grain size and impurity loading. Measurements show that average grain size and impurity content are negatively correlated in ice cores (e.g., Gow and Williamson, 1976; Gow et al., 1997; Lipenkov et al., 1989; Azuma et al., 1999, 2000; Thorsteinsson et al., 1995; Durand et al., 2009). Rempel et al. (2001) highlighted this anticorrelation as consistent with the assumption that "the surface energy of curved interfaces acts to make vein radii uniform (Nye, 1989; Mader, 1992)", reasoning that "variations in [bulk impurity loading] must correlate with changes in the total length of veins per unit sample volume". Effectively, the idea was that by responding to impurity loading, grain sizes could adjust and prevent differences in vein radii from occuring.

By contrast, Ng treats the grain size and impurity loading as unrelated, implying that vein radii are initially in disequilibrium when changes in deposition produce changes in impurity loading, so that the vein radii subsequently evolve towards uniformity. This equilibration of vein radii drives additional computational transport that adjusts the impurity loading, thereby modifying the form of the compositional signals and preventing their displacement relative to the ice.

The analysis by Ng is elegant and clearly presented, and represents a welcome addition to the literature that describes post depositional processes with potential to be relevant to ice core records. To better convey the departure from the earlier efforts to address this particular problem, it would be helpful to clarify the discussion surrounding the different assumptions regarding ice grain size. At present, the reason given by Ng for neglect of the Gibbs-Thomson effect in the Rempel et al. (2001) model is the scaling in the liquidus relation (line 122). However, the original reasoning provided by Rempel et al. (2001, p. 370) emerges from the assumption that an anti-correlation between grain size and impurity loading makes vein radii uniform, as predicted if grain size scales inversely with the square of bulk impurity loading in equation (7) of the current work, thereby rendering the second term in equation (8) spatially uniform as well.

Unfortunately, as noted by Ng around line 239, "reliable grain-size modelling remains out of reach", though impressive improvements have been made in recent years (e.g. Faria et al., 2014; Ng and Jacka, 2014). Nevertheless, no convincing treatment has yet been provided that is quantitatively successful at representing the causal mechanisms that produce the observed anti-correlation between grain size and impurity loading. The current work highlights yet another reason, in addition to more commonly evoked considerations of the effects of grain growth on rheology and fabric development, for the importance of addressing this challenge.

Under the impurity-independent grain size assumption, Ng demonstrates convincingly that short-wavelength compositional variations (with impurities confined entirely to the vein-node network) should not persist over multi-millennial time scales under typical ice core conditions. Clearly, the apparent preservation of such signals deep in the

ice requires further explanation, and Ng provides considerable insight into various potential mechanisms that might resolve this conundrum. Ultimately, two scenarios are championed, with the first involving some mechanism for blocking vein transport, for example with dust particles (e.g. Raymond and Harrison, 1975), and the second relying on impurities being largely confined to the ice matrix or two-grain boundaries (e.g. Eichler et al., 2017). The model of Rempel et al. (2001) shows that a strong coupling between bulk impurity content and grain growth could provide a third mechanism by reducing or eliminating spatial variations in vein radius and greatly damping the influence of the Gibbs-Thomson effect on impurity transport.

Likely some elements of all three scenarios contribute, and efforts to quantify post-depositional changes would benefit from further efforts aimed at unravelling their relative importance. Ng's manuscript represents a valuable, detailed exploration of the end member case in which grain size is independent of bulk impurity content. This clear and cogent analysis has important implications, and it is to be hoped that it will motivate efforts that provide even further constraints on how exactly post depositional compositional migration occurs in nature.

I note that Reviewer 1 has already provided detailed suggestions for minor wording changes and I have nothing significant to add.

**RC3:**

I agree that it would be helpful to signpost a later discussion on the importance of grain-size variations for determining how vein constituents behave. I would also welcome having the ideas surrounding the specific assumption regarding uniformity of vein radii in the Rempel et al. (2001) model attributed to a personal communication. However, I do not think that such a reference is necessary. While it is true that the matter was not discussed at length in that work, the reasoning was explicitly provided by the statement that "as the surface energy of curved interfaces acts to make vein radii uniform, variations in $c_B$ must correlate with changes in the total length of veins per unit sample volume", with the following sentence going on to note the qualitative support provided by observed anti-correlations between bulk impurity content ($c_B$) and grain size.

The twenty years since we completed that work have seen tremendous advances in the community's ability to characterize the physical and chemical characteristics of ice cores at increasingly fine scales. Despite these advances, it is noteworthy that we still lack a quantitative mechanistic understanding for precisely how the anti-correlation that is commonly observed between $c_B$ and grain size develops. Your new model brings welcome attention to the consequences of that important issue. In your comment you: 1. highlight the importance of the detailed form of the anticorrelation for determining the fate of vein constituents, 2. champion the generality of your formulation in being readily adaptable to examine the affects of different grain size evolution laws, 3. assert the need for a mechanistic understanding of the role of impurity loading on grain size, and 4. emphasize the inability of existing theories of grain growth to address this problem. I think we have broad agreement on each of these points, which offer a clear motivation for filling these knowledge gaps and bolstering confidence in the integrity and resolution of these important paleoclimate records.

Beyond the observed fine-scale grain-size variations themselves, an argument in favor of the uniform vein radius assumption employed in the Rempel et al. (2001) treatment is the long-term preservation and apparent fidelity of fine-scale $c_B$ signals recovered from ancient ice. If, as in your model treatment, vein radius evolves to force diffusive impurity redistribution, then your analysis implies that either those deep signals are distorted from their original form, thereby compromising detailed paleoclimate interpretations, or instead their preservation might be attributed to one or both of the mechanisms that you suggest, namely: residence outside of the vein network under much warmer conditions than the eutectic temperatures of their solutions, or blockages that manage somehow to severely restrict vein diffusivity. The observed anti-correlation between impurity content and grain size would in any case remain unexplained. However, should

this problem be addressed, the precise manner in which grain sizes respond to impurity content or perhaps some coincident variable (e.g. impurities on two-grain boundaries) could be accounted for in a refined treatment that extends beyond the locally uniform grain size case that is the focus of the example calculations in your paper. While we're each free to argue over the set of assumptions we feel to be most reasonable, whichever situation actually dominates is not currently known.

---

## Author Response (AR3)

4th March 2021

Dear Editor:

I have undertaken a minor revision of the manuscript by following all of the reviewer's suggestions. My point-by-point response is given below (in black), alongside his comments (coloured blue). Section 3.4 is now more complete with a layperson's explanation and should be more accessible.

Best wishes,
Felix Ng

**Report by Eric Wolff (dated 03-March-2021)**

This is a really impressive and thorough paper. It starts by examining an idea that has certainly caused a lot of interest and concern in the ice core community (cited 87 times), but which seems inconsistent with observed data. Despite this, the mismatch with data has never been explained, and this paper represents a real advance in that it explores the implications of the idea with additional considerations, and starts a discussion of the possible ways of reconciling theory and data. It's quite a tough read for a non-mathematical reader, but with enough simpler explanations that its interesting implications can be understood. I did not review the first version of this paper but I have been asked to look particularly at the new section of text (section 3.4).

Firstly on the inclusion of section 3.4 at all – I think the exchange in the interactive discussion between the author and reviewer 2 was important and it is definitely worthwhile making clear the assumptions that Rempel made about grain size and whether they are reasonable. For this reason I think the first half of section 3.4 should definitely be there, although it may be possible to smooth its edges a little. The second half, where the implications of a non-uniform grain size are explored, is an interesting new angle, and I think is correct. It isn't strictly to the point for this paper and might seem a little distracting, but as it would probably not warrant a separate paper on its own I agree that it should be here, as long as it is clearly explained (which needs some work).

I am therefore suggesting some mainly minor changes (chiefly clarifications) but otherwise I certainly recommend publication.

Thank you for providing this supportive review and sharing your thoughts on the usefulness of Section 3.4 and giving me valuable suggestions below.

In the discussion it was clear that Rempel felt that the justification for neglecting the curvature term in the 2001 paper was clear, while the present author clearly feels it was not. I do not think that the motivation of Rempel et al (2001) on that point is particularly important for readers of this paper and I would suggest some minor wording changes so that this is not a point. In addition this paper should stand without readers needing to look at the discussion, which will just be a distraction. I therefore suggest a rewording of lines 430-434, and that there should be a single and full citation to the discussion without continually referring to it.
Thanks for these reflections. Yes, I am going to follow your advice in the next item to implement those minor wording changes.

Line 431: I suggest "In this connection, in the Interactive Discussions of our manuscript (give proper reference according to TCD style to the necessary discussion comments) it was clarified that Rempel et al. (2001) neglected the Gibbs–Thomson effect from the liquidus relation based on an assumption that the vein radii $r_v$ were spatially uniform – the justification for this being an anticorrelation between mean grain size and impurity loading, which has been observed in ice-core records". I would suggest removing "As explained in RC2" in line 434 as it is obvious this is a continuation of discussing what was in the comment.
Your suggested wording improves the tone and efficiency of that passage. I have implemented these changes. Please see Lines 430–433 and 434.

Line 442 and following, where you discuss the need for dg^2 to be proportional to C_B: there is actually another reason why this is unlikely. The actual freezing point depression, and therefore the equilibrium value of C (the vein concentration) is dependent on the entire mix of chemicals in the liquid phase. The evolution of grain size will also somehow be dependent on different chemicals interacting with grain boundaries. It is vanishingly unlikely that the way the different chemicals combine to control freezing point is the same as the way they control grain size. Thus even if the proportionality was true for one mix of chemicals, it would not be true for a different mixture. This is a kind of extension of your reason (iii). I realise this is a detail and I don't insist that you add it but it might be something else to consider.

Thank you for pointing out this extra reason why the inverse-square relationship is unlikely to hold. Yes, item (iii) in the passage is a suitable place to mention it. I have done this, on Lines 454–456.

For lines 460 onwards, you have not (as in previous sections) given a layperson's explanation of what is occurring here, and I found it hard to work out exactly why this creation of peaks is occurring. I think I got it, so I will give an explanation of my own: if I am right you should include something similar so that those not wanting to follow the maths can still understand the mechanism. I think the argument is:

<<Smaller grain size implies for a given C_B, more vein length and (by equation 4) lower rv. This in turn implies that the Gibbs-Thomson effect in eq 6 is stronger (more freezing point depression) and therefore for a given ice temperature, the solute effect must be lower, ie c must be smaller in the ice with smaller grain size. This leaves a concentration gradient and causes diffusion that raises C_B in the area with small grains at the expense of the surrounding ice >> It might also be worth spelling out that the effect (again from eq 4) is that rv increases, and presumably diffusion only continues until rv has reached the size it is in the surrounding ice (I think eq 4 then tells us that C_B will reach an asymptotic value related to the square of the ratio of d_g inside and outside the perturbation). It would actually be helpful if Fig 12 included the evolution of rv – could this be added?

Thank you for suggesting adding a layperson's explanation and writing a passage to help me. What you described captures the interactions and matches my understanding. Your description is helped by referring to the curves and the intersection point in Figure 2f, which can be used to explain why fluctuation in the grain size perturbs c.  In the revision, following the passage where the mathematical result (32) exposing signal formation is first described, I now give the layperson's explanation in the paragraph on Lines 471–479, by adapting your passage and referring to Figure 2f. Several sentences on Lines 464–470 have been adjusted to coordinate with this change and to control the flow.

     Your final suggestion is to plot r_v to illustrate the layperson's explanation. This is useful, although not so useful to do in Fig. 12, because that simulation run includes vertical compression and changing temperature (with non-zero temperature gradient), so r_v does not become spatially uniform at large time. Only in the simplified situation of the layperson's explanation would r_v evolve to a constant. Plotting r_v in Fig. 12 and describing its complex evolution probably over-complicates Section 3.4. Therefore I have opted to plot r_v in Movie S6, where the reader can see how r_v evolves --- comprehensively, at all times. In the main text, on Line 488–489, I have added a note in brackets to clarify why r_v doesn't become constant. Similarly, I have added panels in Movie S7 (the EPICA run) to display r_v at all times. Accordingly, the Supplementary File and the Data Repository have been updated with new captions and movies.

Finally in the conclusions, the paper contrasts impurities dissolved in the veins, and impurities in the grains or grain boundaries. Remember that they can also be in the veins but not dissolved. As an example at Dome C at -50 degrees, sulphuric acid would be dissolved (well above the eutectic of -70) but NaCl would almost certainly have precipitated out somewhere around -23. This adds a further complication because in my example, NaCl could dissolve back into the veins at the warmer temperatures at depth.

Thank you for pointing out this. In Section 4, I now signpost this additional complication in two places: Lines 585–587 (inside a paragraph about the distribution of impurities in ice) and Lines 602–603 (in a passage highlighting future modelling challenges).

---

## Author Response (AR4)

5th March 2021

Dear Editor:

Thank you for your suggestion on how to format the two passages in Sect. 3.4 of the manuscript:

line 431-432: "In this connection, in the Interactive Discussions of our manuscript (see interactive comment on The Cryosphere Discussions, RC2 (https://doi.org/10.5194/tc-2020-217-RC2 , 2020) and RC3 (https://doi.org/10.5194/tc-2020-217-RC3 , 2020) and associated author comments AC2, AC3 and AC4) it was clarified..."

line 440: "In the Interactive Discussions (see interactive comment on The Cryosphere Discussions, RC2, https://doi.org/10.5194/tc-2020-217-RC2 , 2020), it was suggested.."

I have implemented these changes and am uploading the final set of manuscript files to the submission site.

Kind regards,
Felix Ng
Author of tc-2020-217